# Seismic noise variability as an indicator of urban mobility during COVID-19 pandemic in Santiago Metropolitan Region, Chile

Javier Ojeda[1] and Sergio Ruiz[1]

[1]Departamento de Geofísica, Universidad de Chile, Santiago, Chile

**Correspondence:** Javier Ojeda (jojeda@dgf.uchile.cl)

**Abstract.** On 3 March 2020, the first case of COVID-19 was confirmed in Chile. Since then, the Ministry of Health has imposed mobility restrictions, a global policy implemented to mitigate the propagation of the virus. The national seismic network operating throughout Chile provides an opportunity to monitor the ambient seismic noise (ASN) and determine the effectiveness of public policies imposed to reduce urban mobility in the major cities. Herein, we analyse temporal variations in high-frequency ASN recorded by broadband and strong-motion instruments deployed throughout the main cities of Chile. We focus on the capital, Santiago, a city with more than 7 million inhabitants because it is seismically well-instrumented and has high levels of urban mobility due to work commutes inside the region. We observed strong similarities between anthropogenic seismic noise and human mobility indicators, as shown in the difference between urban and rural amplitudes, long-term variations, and variability due to the COVID-19 outbreak. The same results are observed in other cities such as Iquique, La Serena, and Concepción. Our findings suggest that the initially implemented public health policies and the early deconfinement in mid-April 2020 in the metropolitan region caused an increase in mobility and virus transmission, where the peak in anthropogenic seismic noise coincides with the peak of the effective reproductive number from confirmed positive cases of COVID-19. These results confirm that seismic networks are capable of recording the urban mobility of population within cities, and we show that continuous monitoring of ASN can quantify urban mobility. Finally, we suggest that real-time changes in ASN amplitudes should be considered as part of public health policy in further protocols in Santiago as well as other high-density cities of the world, as has been useful during the recent pandemic.

## 1 Introduction

Since the propagation of the severe acute respiratory syndrome coronavirus 2 (SARS-CoV-2) which causes the coronavirus disease-2019 (COVID-19), countries have used various strategies to reduce the risk of the virus spreading (Walker et al., 2020). In Chile, the first case of COVID-19 was confirmed on 3 March 2020, and then the disease spread rapidly until the number of cases reached a peak on 14 June (Canals et al., 2020). During this first period, the main public health policy addresses the isolation and social distance, including the closure of schools, universities and other educational centres (16 March), national night-time curfew (23 March), and the lockdown of communes. From 19 July 2020, the Chilean government implemented the step-by-step programme, which considers a gradual open of each commune by five phases, based on the monitoring of epidemiological and health system indicators (see Table S1; Tariq et al. (2021)). One of the most affected zones corresponds

to the metropolitan region (hereafter MR) of Chile, which includes the capital, Santiago, a big city with 40% of the Chilean population. According to the 2017 Census (INE, 2017), the population of the MR reached 7,112,808 inhabitants with a density of 461.77 inhabitants per km$^2$, and Santiago had the highest population density of Chile (Figure 1, see Figure S1a).

Ambient seismic noise (ASN) can be attributed to various sources depending on the frequency band analysed. At higher frequencies, above 1 Hz, ASN exhibits daily, weekly and holiday variations linked to human activities (e.g., Bonnefoy-Claudet et al., 2006; Groos and Ritter, 2009; Díaz et al., 2017). This high-frequency band, also called anthropogenic or cultural noise, registered on seismic data is quickly attenuated with distance at the order of a few km (Groos and Ritter, 2009; Boese et al., 2015; Green et al., 2017). The effects of COVID-19 outbreak and global mobility restrictions have created an opportunity to analyse the reduction and temporal variations in high-frequency ASN on seismological networks (e.g., Lecocq et al., 2020), GNSS Observations (Karegar and Kusche, 2020), and fibre-optic distributed acoustic sensing (Lindsey et al., 2020). Previous works in other countries compare the temporal variability between ASN and other observables such as mobility data from cell phone displacements in northern Italy (Poli et al., 2020), Río de Janeiro, Brazil (Dias et al., 2020), Sicily, Italy (Cannata et al., 2021), Auckland, New Zealand (van Wijk et al., 2021), Barcelona, Spain (Diaz et al., 2021), and Querétaro, México (De Plaen et al., 2021). In addition, Xiao et al. (2020) reported cultural noise changes in China, as well as Guenaga et al. (2021) distinguished significant ASN reductions in academic institutions across the United States. These observations reveal the effect of mobility restrictions imprinted on geophysical measurements worldwide, where the pandemic caused one of the quietest recorded periods on Earth with a 50% reduction in seismic noise (Denolle and Nissen-Meyer, 2020; Lecocq et al., 2020). However, a deep analysis of temporal variations in ASN in each city/country could reveal ways to significantly advance the robust monitoring of urban mobility, especially during a pandemic.

Herein, we analyse the effect of temporal variations on ASN using the continuous recording data of the Chilean seismic network operated by the National Seismological Centre (hereafter CSN; Centro Sismológico Nacional; Barrientos (2018)). Our results show the difference between urban and rural ASN amplitudes, and the long-term temporal variations and reduction during holiday seasons as well as those due to the spreading of COVID-19 and mobility restrictions. Additionally, we studied the temporal variations due to the implementation of different public health policies in Chile. We identified an agreement between high-frequency ASN amplitudes and other mobility data within the MR, as well as an increase in epidemiological parameters of virus transmission on the dates of a peak in ASN. Our findings suggest that ASN analysis could be used to infer the dynamics of the population influenced by large and long-scale events in big cities, like a pandemic.

## 2    Data and Methods

### 2.1    Seismic data

Since 2013 the CSN has deployed a large number of seismic stations throughout Chile, most of them in areas outside the main cities where there is a low signal-to-noise ratio to better record the regular seismicity (Barrientos, 2018). The CSN network is formed by $\sim$ 120 broad-band and accelerometer multi-parametric stations deployed at the same location, $\sim$ 300 accelerographs with trigger systems, and a few continuous recording strong-motion instruments. Further information about

geophysical characterization and soil conditions where stations were deployed can be found in Leyton et al. (2018a) and Leyton et al. (2018b). Herein, we focus on stations located inside or close to the principal cities. We analysed the data of fifteen stations placed in the MR (see Figure S1) as well as five stations deployed in some cities within Chile such as Iquique, La Serena, Valparaíso, Concepción, and Puerto Williams (Figure 1, see Table S2). These stations were selected because of their distance from the main cities, high-quality data, and short-time data gaps. We only analysed the vertical component of the data due to the similarities in the results observed using horizontal components (Lecocq et al., 2020). Most of the records correspond to broadband stations, except for the strong motion C.CCSP station placed in San Pedro - Concepción and the strong motion C1.MT18 station located in downtown Santiago (for MT18 station, we analysed both broadband and strong-motion records). For all stations, we processed eleven months of data from 1 December 2019 to 1 October 2020 and three years of data from 1 October 2017 to 1 October 2020, for stations MT09 and MT14 located in the MR.

## 2.2 Seismic noise analyses

To investigate the temporal changes in the seismic signal, we followed the methodology used by Lecocq et al. (2020) considering continuous data. We computed a daily power spectral density in 30-min windows, where each windowed time series was calculated using Welch's method (Welch, 1967), therefore, the windowed segments were converted to periodograms. We estimated the displacement spectral power, from which we calculate the root-mean-square (RMS) of the time-domain displacement using a bandpass filter. To better understand the effects of the chosen corner frequency applied in the bandpass filter, we tested the temporal changes in the normalised seismic RMS amplitude at the MT18 station (Figure 2). In the first order, the temporal changes are similar between the different corner frequencies applied, excluding the lower frequency bands such as 0.1-1 Hz, 1-3 Hz, and 3-5 Hz. At frequencies > 1 Hz we can avoid records of microseism and observe important temporal changes in the normalised displacement. Consequently, we decided to filter the data between 4 and 14 Hz to obtain the RMS of the time-domain displacement or high-frequency ASN (Lecocq et al., 2020).

We calculated the RMS displacement variability for broadband and strong-motion records. Figure 3 shows the temporal changes in ASN amplitudes that are comparable between both instruments at station MT18. The median day-time amplitudes between 5h and 22h local time obtained from the seismometer and the accelerometer exhibit similar trends and behaviour. To analyse the seismic effects of lockdown in Santiago City, we calculated 24-h clock plots for station MT18 through which we observed the average displacement variation for weekdays and weekends for the period before Lockdown 1 (Figure 4a) and after Lockdown 1 (Figure 4b). In addition, the displacement noise evolution is shown in an hourly grid representation from January 2020 to August 2020 (Figure 4c).

## 2.3 Other observables: Epidemiological and mobility data

Our study integrates epidemiological data available on the website of the Chilean Ministry of Science. One of the primary indicators of the spreading of viruses and contagion dynamics is the estimation of the effective reproductive number (hereafter Re) from confirmed positive cases of COVID-19 since the date of the beginning of symptoms. The Re indicator is defined as the actual average number of secondary cases generated by a primary case during the epidemic outbreak (Caicedo-Ochoa

et al., 2020; Tariq et al., 2021), their estimation is helpful to the assessment of public policies, to estimate population immunity, to monitor near real-time changes in transmission of the viruses over time, among others (Gostic et al., 2020). To control an epidemic outbreak, the Re indicator needs to be reduced below one (Riley et al., 2003). Herein, we used the estimation provided by ICOVID Chile (2020) who described the function Re depending on the proportion of susceptible individuals to be infected, a transmission coefficient and the infectious life expectancy. In other words, the Re accounts for the coefficient between the new infections and the recovery rates plus mortality rates (Contreras et al., 2020). ICOVID Chile (2020) used the method proposed by Cori et al. (2013) to monitor Re in real-time, modelling the transmission like a Poisson process calculated on the basis of the last seven days. We considered only the Re median and 95% credible interval estimated for the urban area in the MR, according to the data given by the Health Service of Santiago City.

The mobility data we analysed is provided by Apple mobile-phone locations in Santiago City, which corresponds to the percentage of change in the public's walking and driving in relation to a baseline value from 13 January (Apple, 2020). Moreover, we used the public transport transactions provided by the Chilean Ministry of Transport and the Instituto de Sistemas Complejos de Ingeniería (Ministry of Science, 2020). They account for the total number of validations using the public transportation card in the MR. This mobility card is the only system to make transactions in public transport. Finally, we utilised local flight data provided by the Civil Aeronautics Board, which corresponds to the total number of passengers that departed from Santiago City or arrived there on national flights (Ministry of Science, 2020) to the Airport "Arturo Merino Benítez" located approximately 5 km western to station MT05 (see Figure S1c).

## 3 Results

### 3.1 Lockdown, curfew and ASN amplitudes

We analysed the seismic effect caused by the first lockdown in Santiago City using the 24-h clock plots in station MT18 (Figure 4a, 4b). Although we observed a gradual reduction in ASN amplitudes on weekdays due to the day-cares, schools and universities near the station closed (16 March), we also notice a strong reduction on weekends, especially between 11h and 19h local time. Figure S1b shows the area close to MT18 in which we can distinguish the hippodrome "Club Hípico de Santiago" and the O'Higgins Park. The highest ASN amplitudes observed on Saturday before Lockdown 1 (Figure 4a) is explained by the activities of the hippodrome on Saturdays (and some Thursdays during January-February). The hippodrome closed on 21 March 2021, which is in agreement with the decrease in the ASN amplitudes observed after Lockdown 1 (Figure 4b).

We also distinguish the lockdown effect in the hourly grid representation (Figure 4c). The large ASN amplitudes observed during holidays are associated with near activities in both hippodrome and O'Higgins Park, which only persist on weekends during March. After the implementation of Lockdown 1, the ASN amplitudes drop, especially on weekends. Moreover, we observed a systematic behaviour of lower ASN amplitudes between 22h and 5h local time due to the overnight curfew implemented at the same hours, imposed from Lockdown 1 and remain during the full time-window studied.

## 3.2 ASN variability between urban and rural areas in the Metropolitan region

Figure 5 shows the ASN variability over three years for two representative stations in MR, MT14 in Las Condes, and MT09 located on an almost unpopulated area, near the town of Talagante. High-frequency ASN exhibits higher amplitude in urban stations than rural environments. MT14 presents larger temporal changes, for example during the summer holiday season (January-February). The same trends are observed in the three analysed years 2018, 2019, and 2020, and the lockdown restrictions are visible with clear changes in the mean of ASN. In contrast, the MT09 station located in a rural environment presents lower ASN amplitudes with few and low peaks due to local activity in the area.

Figure 6 shows the ASN variability in the six stations inside the urban ratio in the MR. In ten months of data, we can see the temporal changes in high-frequency ASN. MR had different lockdown protocols depending on each municipality (see Table S2); however, most of the changes started on 16 March 2020, where schools, universities, and institutes decided to shut down and stopped their activities to mitigate the spread of COVID-19. The first lockdown was in downtown Santiago (MT18, Figure 6b) and Las Condes (MT14, MT16, Figures 6c, 6d) on 26 March 2020, twenty-three days after the first case of positive COVID-19 and was lifted on 13 April (downtown Santiago, MT18) and 16 April (Las Condes, MT14, MT16). After that, ASN amplitudes increased until the second lockdown on 5 May (downtown Santiago, MT18) and 15 May (Las Condes, MT14, MT16) where immediately ASN amplitude decreased. These temporal restrictions were well recorded by seismic stations installed in urban areas and are useful for analysing urban mobility. Other sectors from MR such as Renca (MT05, Figure 6a), Peñalolén (MT03, Figure 6e), and Puente Alto (MT15, Figure 6f) also show strong variations in ASN because of lockdown restrictions. Concerning the ASN amplitude variability, we observed that the quieter stations in the urban area of MR correspond to MT05, MT14, MT16 and MT03, stations that are located over hills, unlike the MT18 and MT15 stations which are deployed in the valley. Despite the ASN present an average amplitude difference between each station, the temporal variations can be observed within Santiago City. In contrast, the high-frequency ASN estimated for rural stations show lower amplitudes and without conspicuous evidence of temporal variations due to mobility restrictions (see Figure S2); as we mentioned previously, located peaks are related to local activity in those areas and do not seem to follow a behaviour related to the anthropogenic ASN. Notice that the rural stations analysed are deployed within the MR at a distance of about 15 km to 60 km from the stations installed in urban areas (see Figure S1).

## 3.3 Mobility restrictions, and epidemiological parameters linked with seismic noise amplitudes

We analysed the temporal variability of high-frequency ASN for station MT14, located in Las Condes municipality within Santiago because this station lacked data gaps during 2020 and the area implemented a diversity of public policies for mitigating the effects of the pandemic. Figure 7 shows a matching pattern between the temporal changes in ASN amplitudes and the effective reproductive number (Re), especially from March to July, where we observed a peak in the number of positive COVID-19 cases (Canals et al., 2020). The peak for the Re indicator occurred after the end of Lockdown 1 and before Lockdown 2, a period in which the ASN amplitudes also increased due to the early deconfinement promoted by a "safe return" to normal activities (Tariq et al., 2021). This observation is supported by mobility data such as Apple driving and walking data (see

Figure S4), the gradual increase in public transport transactions, and a gradual increase in the flight arrivals and departures from Santigo city (see Figure S5).

After Lockdown 2, in mid-July, the Chilean government proposed the step-by-step programme to mitigate the propagation of SARS-CoV-2 virus towards a gradual re-opening and increase mobility in different counties as a public health policy (see Table S1). The programme considers five phases, where the citizens progressively increase their mobility, and the advance or retreat of these phases is related to the epidemiological situation of each municipality. Las Condes (MT14) was the first territory in the MR in which the programme changed from Phase 1 to Phases 2 and 3. These mobility conditions explain the sharp increase in high-frequency ASN amplitudes after 28 July (Phase 2) and 2 September (Phase 3) (Figure 7). These periods also correlate with the strong increase in the Apple mobility data. The Re indicator presents a gradual increase in this period and seems to oscillate around one, similar to the time-window between May and June.

## 3.4 Situation in other places along Chile

High-frequency ASN changes were also recorded in other cities along Chile (Figure 1). The variability in ASN amplitudes was observed in other populated cities of northern Chile, such as Iquique (TA02, see Figure S3a) and La Serena (CO05, see Figure S3b). In Iquique, the high-frequency ASN decreased after school closures until the end of April 2020, after which ASN increased to average levels. With the increase in the number of positive COVID-19 cases, the government implemented the first lockdown in this city on 15 May, dates in which we observed an instantaneous drop in ASN amplitude which gradually recovers until normal seismic noise levels at the end of September. In La Serena, the trends were similar, with a local ASN decrease on 16 March that ceased at the end of April. However, data gaps at station CO05 in La Serena do not allow further analysis. The Valparaíso station, along the coast of Central Chile (VA01, see Figure S3c), showed high variation due to the influence of the ocean (Cessaro, 1994; Ardhuin et al., 2011), and it is difficult to highlight the change in ASN associated with anthropogenic noise. In Central-South Chile in San Pedro, Concepción (CCSP, see Figure S3d), we observed a strong reduction in ASN after school closures and the implementation of the first lockdown. The ASN amplitudes in Concepción are at least 30 nm noisier than Santiago (MT18), which could be explained by their location on residential areas, but also the different soil conditions where the stations were installed. Finally, in Puerto Williams (MG01, see Figure S3e), a small town at the end of South America, temporal changes in ASN were also recorded, and this reduction coincided with the lockdown period and restriction of mobility and human activities. Nevertheless, this station shows a strange pattern before Lockdown. The first one corresponds to a high drop in mid-December until the first days of January associated with holidays festivities (Christmas and New Year Day). The second pattern observed is the temporal variability that could be associated with the activity of the airfield near the site where the station is operating. Unfortunately, we do not have access to the aeroplane activity in those weeks to support our assumption.

## 4    Discussion

During 2020, the MR exhibited temporal changes in the amplitude of high-frequency ASN at all the stations near urban areas. However, we also identified temporal changes in the long-term variability of ASN, especially for stations deployed in urban areas than rural areas. Urban areas show larger RMS displacement amplitudes than rural areas located far from cities. We suggest that the variability in ASN amplitudes is a consequence of anthropogenic activities that affect higher frequencies (Díaz et al., 2017).

Since the implementation of lockdown measures in the MR, we observed a strong reduction in the amplitude of ASN similar to that observed by other authors in China, Italy, Brazil, and worldwide (e.g., Dias et al., 2020; Lecocq et al., 2020; Poli et al., 2020; Xiao et al., 2020). Due to the frequency band applied to obtain the RMS displacement, this ASN reduction indicates an intrinsic anthropogenic origin due to the variations in population mobility within cities. We described the temporal patterns and variations between weekdays and weekends, especially in station MT18 in the downtown Santiago, located near an amusement park with high levels of activity during the weekends observed in the seismic data before Lockdown 1 (Figures 3 and 4). After Lockdown 1, the activity on weekends abruptly decreased, whereas the activity during weekdays had a gradual reduction. As a public health policy, from 23 March the government imposed an overnight curfew from 22h to 5h. The impact of this measure was recorded by most of the stations in urban areas, in which the ASN amplitudes decreased abruptly during the study period.

Temporal variations in ASN amplitudes coincide with the dates of changes in public health policies; variations due to lockdown and different phases of deconfinement are observed in the high-frequency ASN amplitudes. One of the most interesting examples in the MR corresponds to station MT14 in Las Condes (Figure 6c). Here, we observed a remarkable increase in ASN amplitudes between 16 April and 15 May, a period in which the government ended the first lockdown and started deconfinement in the eastern MR, raising the individual displacement of the population and urban mobility. Furthermore, during this period we observed a higher Re value, an epidemiological indicator that represents the velocity of propagation of viruses and increases with the transmission and overload of health systems (Caicedo-Ochoa et al., 2020). The mobility data gathered from Apple devices, public transport transactions, and flights (see Figures S4, S5) confirm the increase of urban mobility in the period analysed. Also, Cuadrado et al. (2020) suggest that small-area lockdowns and reductions in mobility can reduce the transmission of the virus but their impact was smaller than the early closures of schools, universities and other educational centres. These observations indicate that ASN can act as an indicator of urban mobility and can be applied to monitor the dynamics of the populations within cities. The Re parameter increased during the periods of higher human activity ("safe return" from LD1) and, therefore, ASN amplitudes increased in a period without strict management of the pandemic. The early removal of lockdown protocols to reopening the economy resulted in a new wave of infections and an exponential increase in the number of positive COVID-19 cases (Tariq et al., 2021), with a peak on 14 June and a 95% intensive care unit bed occupation (Canals et al., 2020).

The implementation of new health public policies like the step-by-step programme and their five phases since mid-July was accompanied by a gradual decrease in the Re parameter, which oscillated around one since those dates. Although the ASN amplitudes increased after Phase 2 and Phase 3 of deconfinement in eastern MR, the Re parameter was not linked, likely

indicating better management of the epidemic outbreak with the broad-scale social distancing interventions applied in MR (Tariq et al., 2021). However, according to Canals et al. (2020), a relaxing of these interventions could rise the infections and saturate the health systems.

The matching pattern between the mobility data, the Re indicator and the high-frequency ASN amplitudes is well established for the station MT14 located in Las Condes. Nonetheless, this did not occur with other stations in urban areas such as MT18

placed in downtown Santiago. This observation can be further explained due to the heterogeneity in policy effectiveness against the COVID-19 spread in MR. Bennett (2021) showed that social distancing, quarantines and testing availability are affected by geographical and socioeconomic factors, in which the lockdowns have been more effective in high-income zones (such as Las Condes) rather than lower-income zones (such as the others station analysed in the MR urban area). Furthermore, the people living in high-income zones can reduce their mobility by around 60% while people in low-income zones only reduce their

mobility by around 20% during lockdown (Carranza et al., 2020).

Finally, we show five examples within Chile of different temporal changes in high-frequency ASN amplitudes, in which the majority show similar patterns of gradual decreases in RMS displacements due to lockdowns or mobility restrictions. The cities like Iquique, La Serena, and Concepción are highly populated regions in Chile, whereas the temporal variations on ASN are comparable with other observations in the MR urban area. Valparaíso, the second most populated region only presents ASN

dominated by the ocean in the frequency band analysed. In Puerto Williams, a small town in the extreme of South America, the decrease in ASN can be explained by the flight operation and restrictions, due to the closeness of the seismic stations and the local airfield. Nonetheless, further analyses including other sources of mobility data and epidemiological indicators are needed.

## 5 Conclusions

Although seismic data have been used to study wave propagation from seismic sources or to monitor different geological

phenomena, temporal changes in the amplitudes of ASN have also been used to monitor population mobility within cities. We demonstrate that we can analyse the dynamics of the population and urban mobility through high-frequency ASN variations. The public health policies implemented to mitigate the spread of COVID-19 had an important influence on citizens mobility; lockdown, curfew, dynamic quarantines, and other restrictions affected the ASN amplitudes recorded by seismological stations. These temporal variations were also observed during the periods of lockdown and gradual deconfinement in the MR of Chile,

in which a sharp decrease in ASN amplitudes was observed in the station within urban areas, in contrast to those in rural areas. Other mobility data were also compared with temporal variations in ASN amplitudes, showing good agreement with data from Apple devices, public transport transactions and national flights. In mid-April, during the rise of the number of positive COVID-19 cases worldwide, the government declared the end of the first lockdown in eastern Santiago. The public policy measure was followed by an increase in the citizen's mobility, a behaviour that was recorded by seismic stations. Most interestingly, during

that period the effective reproductive number increased until it reached a maximum at the beginning of May. Thereafter, the Chilean government declared a second lockdown due to the increase in the number of positive COVID-19 cases in the MR. This observation suggests that the early end of the first lockdown in Santiago, while the pandemic situation was not controlled,

increased the mobility within cities and promoted virus transmission in those weeks, reaching a maximum number of positive cases in mid-June in the MR. Finally, we showed the possibility of real-time monitoring of the population mobility dynamics through time variations of high-frequency ASN amplitudes in both broadband and accelerometer stations. These seismometers are typically used for the management of seismological networks in urban areas; however, recent studies show the potential opportunity to use them as a tool to teach seismology to school students (e.g., Subedi et al., 2020) and increase the interest of society toward Earth Sciences (e.g., Diaz et al., 2020). The analysis of this study could be implemented in other high-density cities of the world with important implications in our society directly or indirectly affected by the COVID-19 pandemic.

*Code availability.* All codes to analyse the seismic data and reproducibility of high-frequency ASN are available in Lecocq et al. (2020).

*Data availability.* The data shown in this study are available requesting from the authors. Seismic data from CSN, networks C and C1 (Universidad de Chile, 2013) are publicly available through the Incorporated Research Institutions for Seismology Data Management Center (IRIS-DMC, last access: November 6th, 2020), Apple mobility data is available at https://covid19.apple.com/mobility (last access: October 2nd, 2020). Public transport transactions and flight data are delivered, and freely available at https://github.com/MinCiencia/Datos-COVID19 (last access: November 11th, 2020). Effective reproductive number estimation and other epidemiological parameters are available at https://www.icovidchile.cl/ (last access: October 2nd). Some figures were made using the Generic Mapping Tools (GMT) software version 5.3.1 (Wessel et al., 2013).

*Author contributions.* JO and SR wrote the manuscript and JO did the seismic noise analysis.

*Competing interests.* The authors declare that they have no conflict of interest.

*Acknowledgements.* This work was partially funded by the Agencia Nacional de Investigación y Desarrollo/Fondo Nacional de Desarrollo Científico y Tecnológico (ANID/FONDECYT) project number 1200779, and Programa de Riesgo Sísmico (Actividades de Interés Nacional [AIN], Universidad de Chile). JO acknowledges support from the ANID (Scholarship ANID-PFCHA/Doctorado Nacional/2020-21200903). We especially thank the operators of the CSN of the Universidad de Chile, who despite the pandemic have continued monitoring and maintaining the Chilean Seismological Network. We would like to thank the fruitful comments of two anonymous reviewers who helped to improve our manuscript.

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

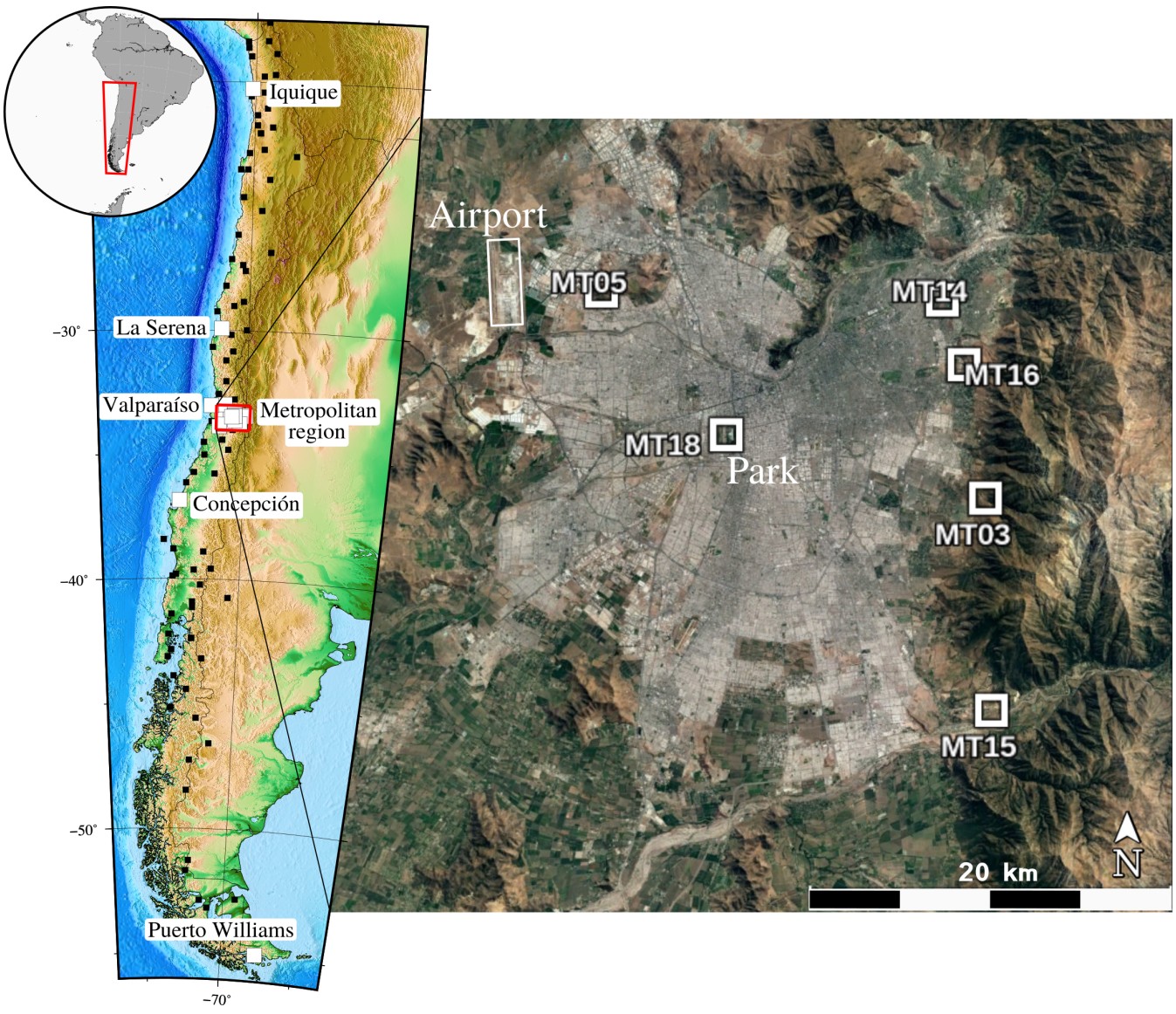

**Figure 1.** Location of the seismic stations of Chilean seismological network. Inset map shows the study location with reference to South America. The small black square corresponds to the broadband stations managed by CSN. White squares correspond to the five stations analysed in Iquique, La Serena, Valparaíso, Concepción, and Puerto Williams. The red rectangle shows the Metropolitan region with six stations located in urban areas (squares with white contours). We showed the location of the Airport and O'Higgins Park. Photos from ©Google Earth.

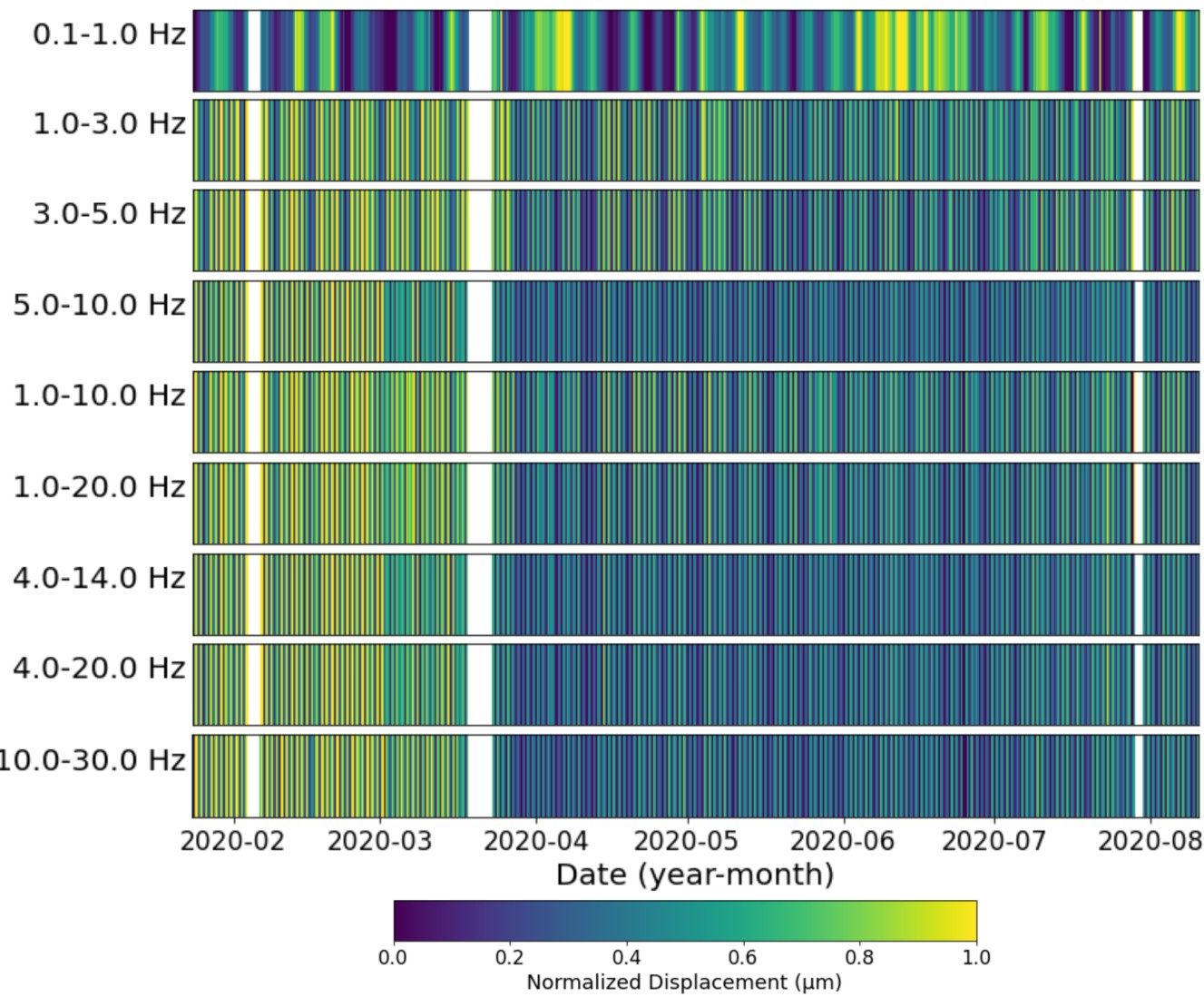

**Figure 2.** Normalised RMS amplitude at the vertical component of station MT18 in downtown Santiago. We tested nine bandpass filters between 0.1 to 30 Hz (see y-axis). White space corresponds to data gaps.

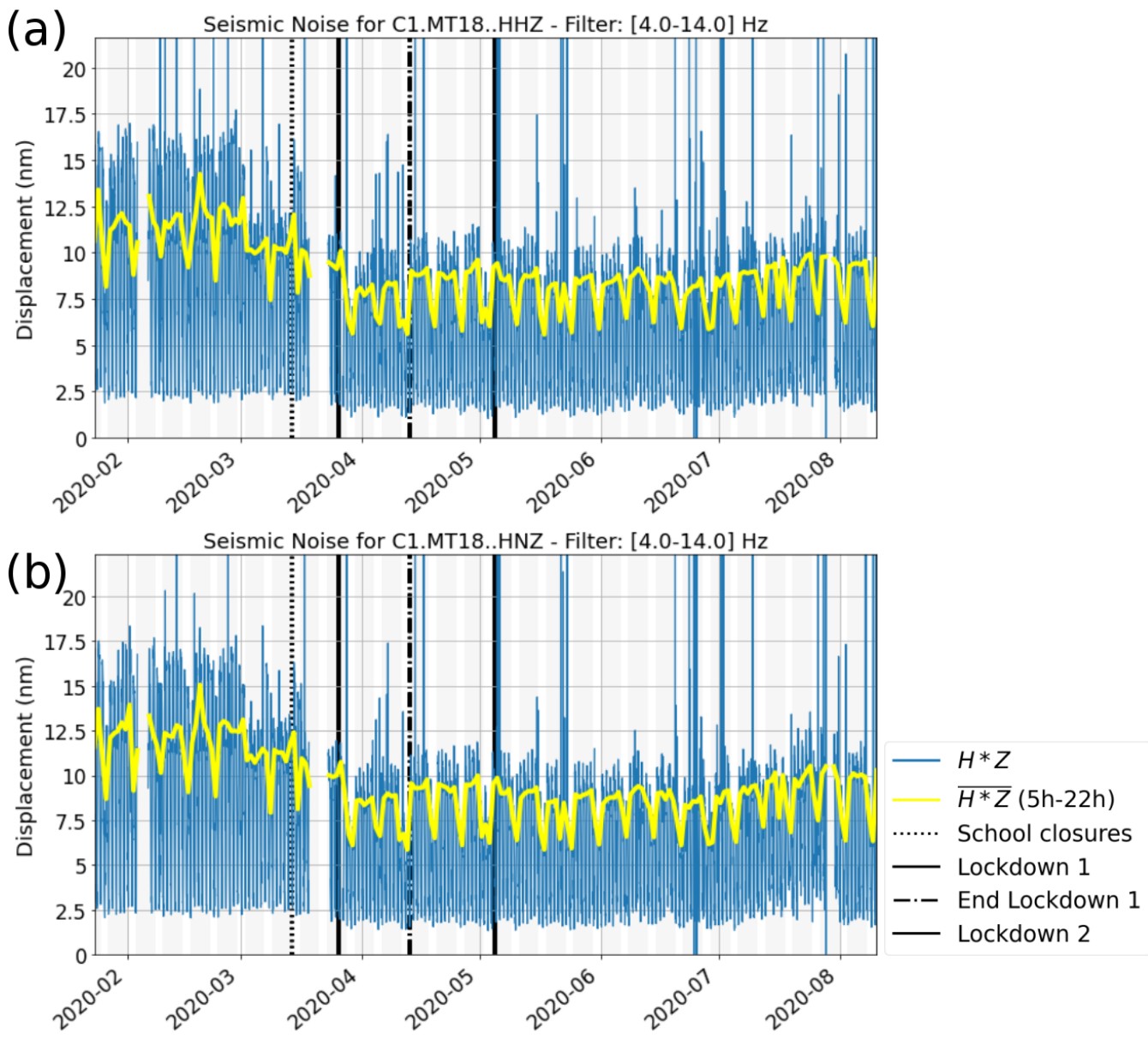

**Figure 3.** Comparison between (a) broadband and (b) strong-motion seismic noise amplitudes for station MT18 in downtown Santiago. The blue line corresponds to the RMS amplitude time series of the vertical component, filtered between 4-14 Hz, and the yellow line corresponds to median day-time, between 5h-22h local time. Gaps correspond to periods for which seismic data are unavailable. The vertical black lines indicate the time of public restrictions implemented in Santiago. The grey and white background corresponds to weekdays and weekends, respectively. Key legend H*Z can be applied for broadband (HHZ) and strong-motion (HNZ) seismic data.

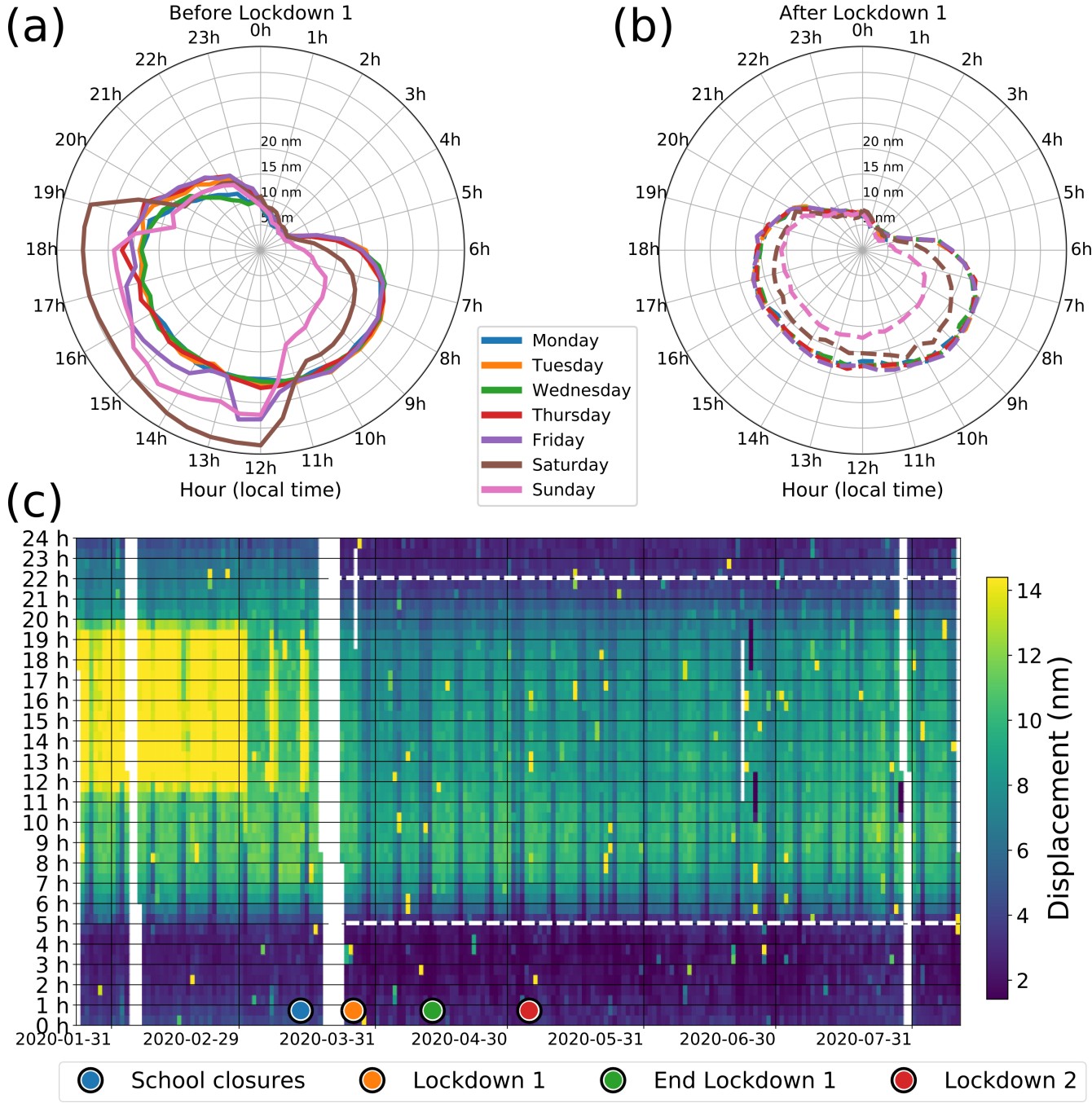

**Figure 4.** Analysis of station MT18 in downtown Santiago. (a, b) Clock plots showing an average of the displacement variability for each day of the week for the period (a) before Lockdown 1 (period 23 Jan 2020 - 25 Mar 2020) and (b) after Lockdown 1 (period 26 Mar 2020 - 10 Aug 2020) at station MT18 in downtown Santiago. (c) Displacement noise evolution is shown in an hourly grid representation. Gaps correspond to periods for which seismic data are unavailable. Blue, orange, green, and red circles below represent the time of public restrictions implemented in downtown Santiago. Horizontal dashed white lines show the curfew period imposed between 22 h - 5 h local time.

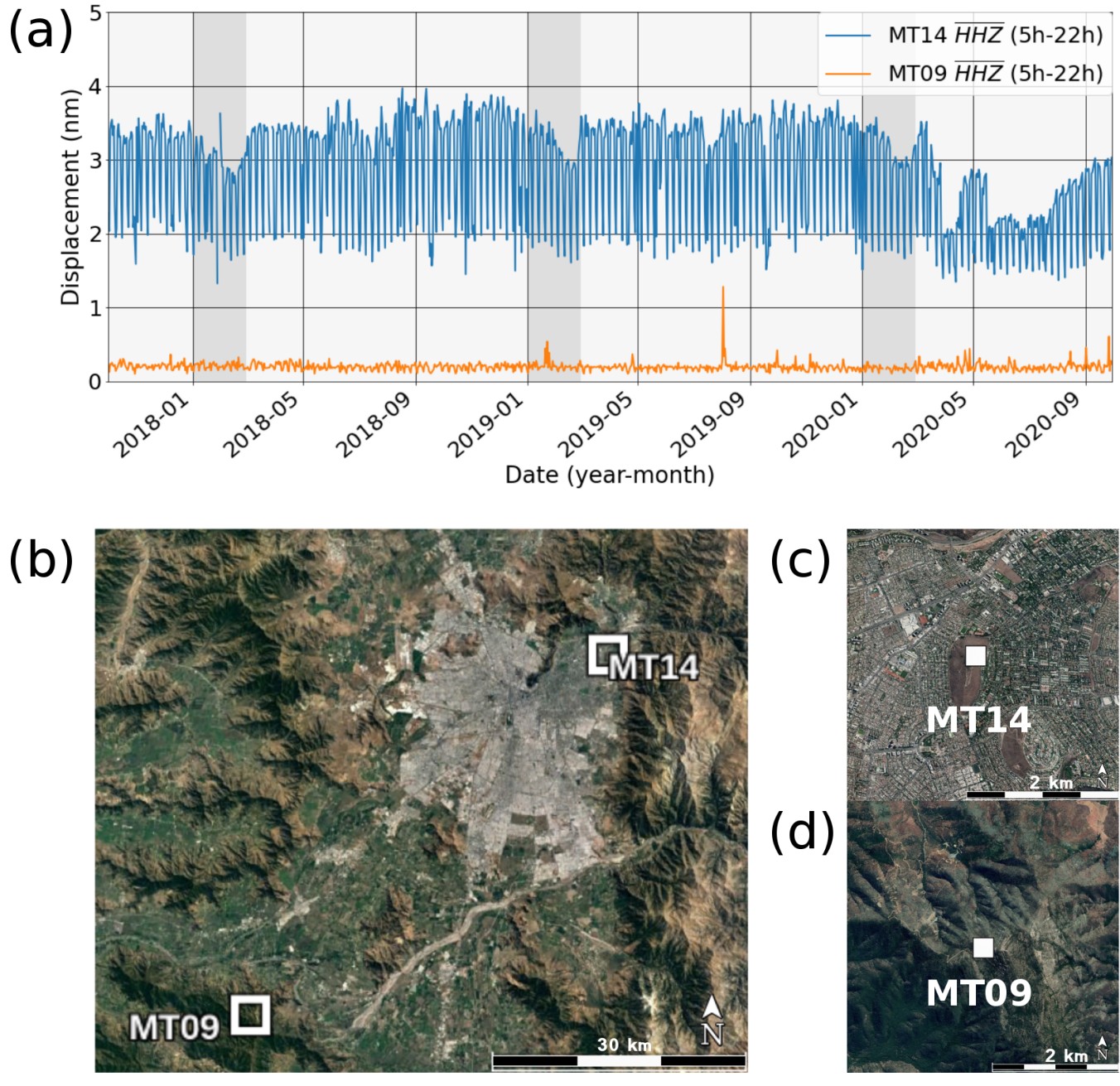

**Figure 5.** Displacement noise evolution at urban and rural stations in the Metropolitan region for the period 1 Oct 2017 – 1 Oct 2020. (a) Long-term noise evolution in station MT14 (blue line) and MT09 (orange line). The grey background correspond to summer vacations. (b) Map with the relative position of each station in the Metropolitan region, and the near 2 km distance from stations (c) MT14, and (d) MT09. Photos from ©Google Earth.

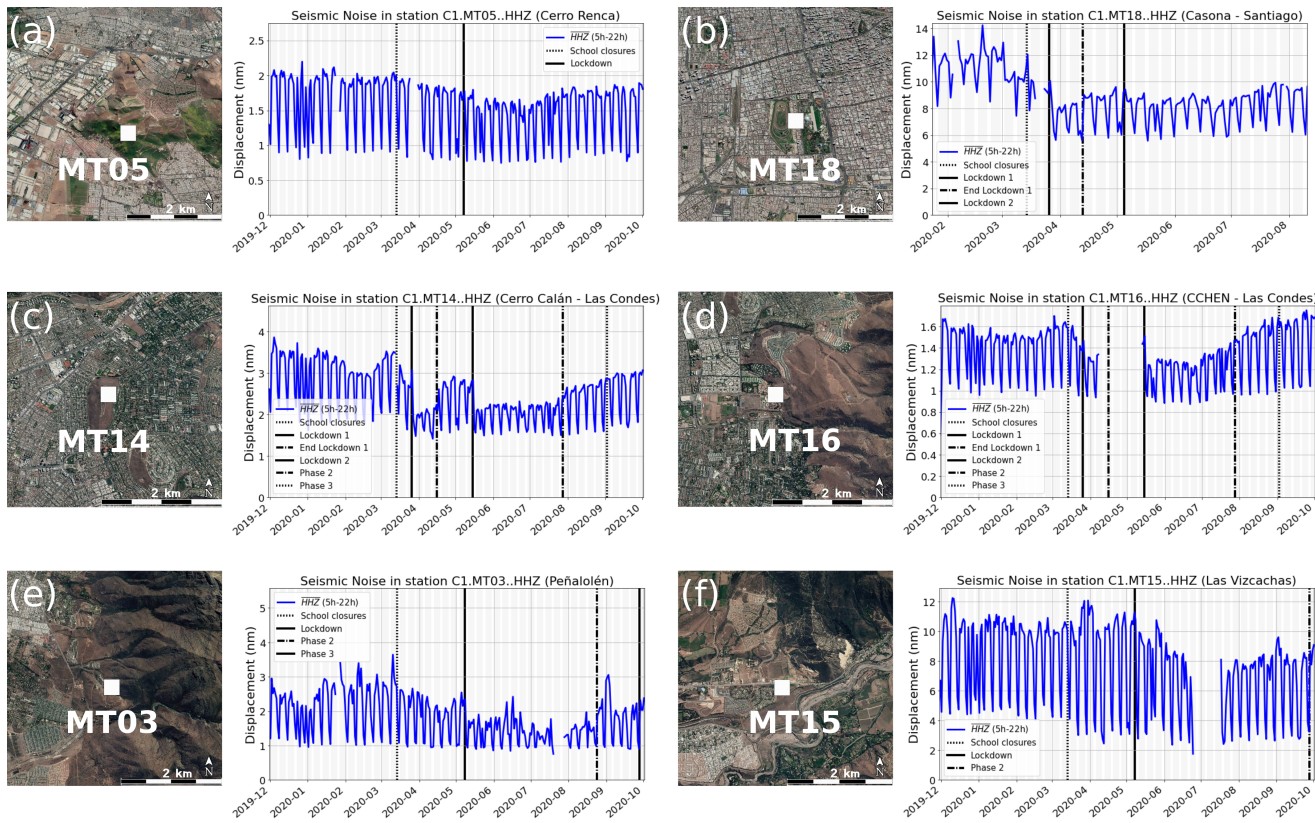

**Figure 6.** Changes in high-frequency seismic ambient noise amplitudes at stations (a) MT05, (b) MT18, (c) MT14, (d) MT16, (e) MT03, (f) MT15. The vertical black lines indicate the time of public restrictions implemented in the Metropolitan region. Each municipality inside the region had different lockdown periods or phases, despite the proximity between them. Photos from ©Google Earth.

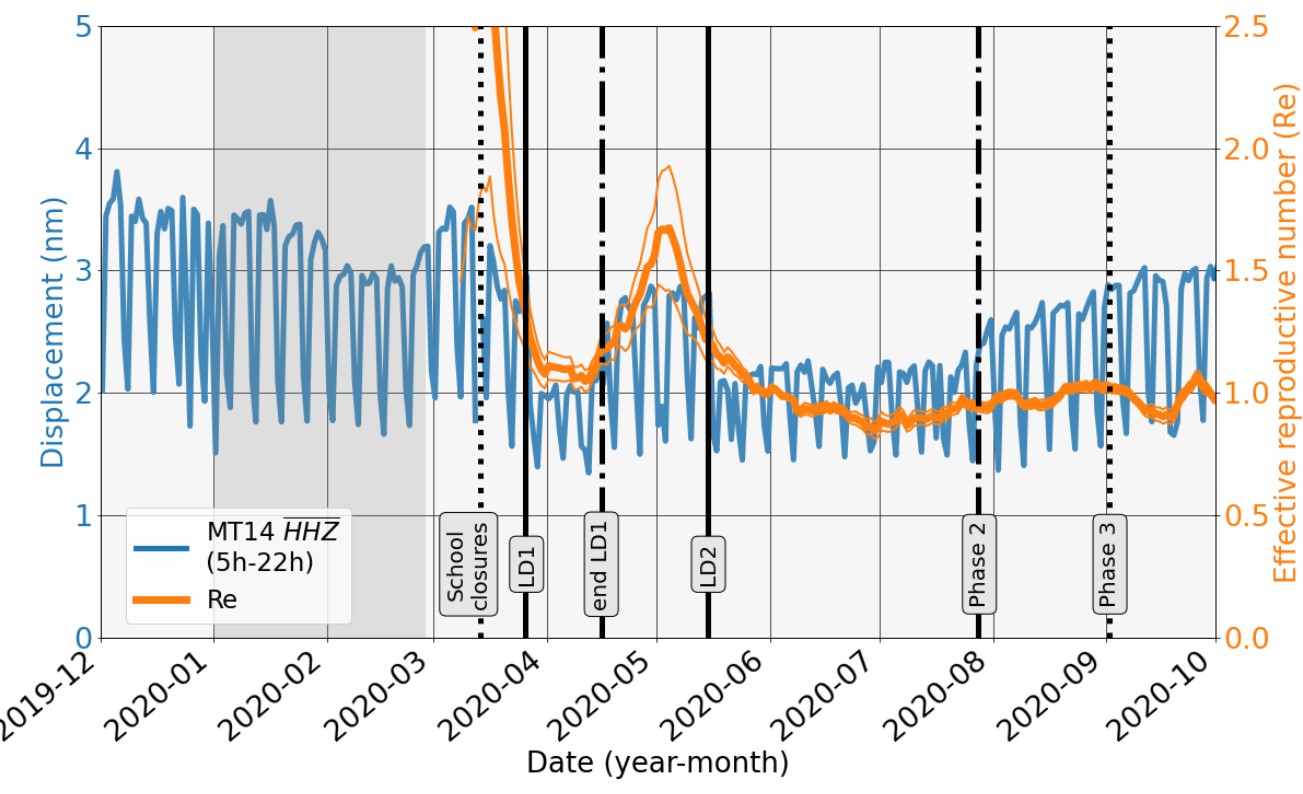

**Figure 7.** Comparison between noise evolution of station MT14 (blue line), the epidemiological factor of the effective reproductive number (Re) over a time is shown (orange line) with their 95% confidence intervals (thin orange lines). The vertical black lines indicate the time of public restrictions implemented in Las Condes. The grey background between January and March corresponds to the Summer holiday season in Chile.