# Peer review of "Seismic noise variability as an indicator of urban mobility during COVID-19 pandemic in Santiago Metropolitan Region, Chile"

_Solid Earth, 2020_

## Referee Comment (RC1) · Anonymous Referee #1 · 15 Jan 2021

Summary

This paper looks at the links between seismic ambient noise (ASN) recorded in Chile and the implementation and reduction of mobility restrictions imposed due to the Covid-19 pandemic. The work looks at both temporal variations and differences within Chile. An interesting link between the ASN and the "R" value is suggested; it will be interesting to understand how this association may be manifested in other countries with different working/commuting patterns. The strength of this link may be over-stated in this pre-print, but it is worth pursuing, as the authors point out. I raise only queries and technical points for the authors to address, and look forward to them making the needed minor

adjustments.

Queries raised

You link ASN and Re. For the readers who are unused to looking at Re, could you indicate whether the Re timeseries you plot are thought to be a lagging indicator (ie does the value calculated refer to infection on a particular date, or does it refer to detection on a particular date and therefore lag when infection actually took place.)

Following on from the previous point: a little more description of the Re calculations would be helpful. For example, does the Re cover all of the MR, or a district of the MR containing seismic station MT14. This kind of information would be useful to consider in the context of the limited reach of ASN noted in line (line 46) and the mention of small-area lockdown in line 230.

You plot changes in ASN and mobility in figure A4. What are these changes relative to? (ie what is 100%?).

Figure 4 shows the MT18 data. Why does it end in August when you have data up to October?

Lines 108-112 are results, not methods. Consider moving them to the appropriate section.

Was lockdown 2 ever lifted? What are phases 2 and 3? it might be good to have a brief paragraph explaining these (somewhere before the results section would seem to be appropriate). Some of this material is already present in lines 176-184.

You report a "strong correlation" (line 167) between Re and ASN at station MT14. Correlation is often used in a mathematical sense. Do you have a mathematical relationship in mind here, or are you looking more at matching patterns?

The link between Re and ASN is stronger before 'phase 2' than after it. Is there any reason for this? Might there be some ASN generating activities which are not linked to
changes in Re? (I am not an epidemiologist, and this is not an epidemiological paper, but at least acknowledging that the relationship between Re and ASN changes seems to be appropriate).

The paper already mentioned other work in other countries, but I would appreciate a brief paragraph which let me know if the links between ASN and other observables are comparable to, or stronger or weaker than, other metropolitan areas. Does the MR look like Barcelona or Mexico City or Rio de Janeiro or Auckland (this may be beyond the scope of the paper and in that case the authors should feel free to ignore this comment).

Technical corrections

Many of these are linguistic suggestions, and the authors may choose to ignore the suggestions – they're not required changes.

* The first sentence (line 34-35) would benefit from a reference from the scientific policy literature.

* Line 40 – km2 → km2

* Line 46 – anthropic → anthropogenic (we're making the noise).

* Line 92 – to better understand the effects of the chosen corner frequency?

* Lines 108-110 – be clearer about the time windows over which the 'gradual' reduction happens, and when the changes cease.

* Line 112 – the noise doesn't go back after lockdown 1 lifts – can you comment on this?

* Line 124 – "Related to mobility data, we analysed" → "the mobility data we analysed is"

* Line 125 – What actually is

* Line 128 – could you explain what a mobility card is?

* Line 158 – do you know if the local activity is more likely to be anthropogenic or seismogenic, or is it hard to tell from the data available?

*Line 167 – different to what? (a range of different responses?)

* Line 194 – might benefit from a reference to oceanic seismic noise for the interested reader?

* Line 280 – which network code is appropriate? And is there a doi for the seismic network which could be cited here (this will help the network operators if they are looking to see which papers use their data).

* Line 322 – is there a volume + page range for Caicedo-Ochoa et al?

* Line 331 – doi or link missing for Cuadrado et al

* Line 367 – add the rest of the author list? Not sure what SE editorial policy is.

* Line 415 – square → squares

* Line 428 – you have an otherwise un-defined term in the key (H*Z). I assume it's because you've got two different components used at this location, but it's not explained anywhere. Consider relabelling/explaining.

* Line 435 – the icons for the school closures, and lockdowns 1 & 2 are really hard to distinguish between.Maybe color more of the icons? Also, is the after-lockdown 1 clock plot (b) for this whole time window, or just until the end of lockdown 1? Please clarify.

* Line 446 – not sure what ratio means here?

Lines 76, 78 + others + 471 – localized (or localised) → located

* Line 24-25 – "Finally, we suggest to consider monitoring in real time the changes in ASN amplitudes to beincluded in the public policies" think about changing to something like "Finally, we suggest that real-time monitoring of changes in ASN amplitudes should

be considered as part of public health monitoring".

---

## Referee Comment (RC2) · Anonymous Referee #2 · 23 Jan 2021

The paper is well written and straightforward. Although several cases of the noise level decrease due to lockdowns, curfews, quarantines in different places have been documented in several countries during last year, it is welcome to have another example from a big city that correlates the decrease with some social index. I recommend the paper for publication after some review.

Abstract: Please, consider to mention in the abstract the results with other stations in Chile (section 3.3)

Line 24: real time to real-time

Line 25 : high density to High-density

[Figure]

Line 35: risk of the spreading of the virus to risk of the virus spreading.

Line 65: Please, consider to include large and long-scale events in big cities. I believe that this is important to stress.

Line 76: Is there any particular reason for the installation of the stations in the capital?

Line 76: Please, provide further information on the conditions where those five stations are installed. Any special treatment because they are inside the city?

Line 80: Although you are using the horizontal components to see the noise decrease, I recommend to perform the noise polarity analyses to check the direction(s) of the noise origin(s). I believe it is important to check with the direction of the park, airport.

Line 83: For all stations we processed to For all stations, we processed

Line 93: Normalized in relation to what? Normalized individually or among then?

Line 103-104: The median day-time amplitudes between 5h and 22h local time obtained from the seismometer and the accelerometer exhibits similar trends and behaviour to The median day-time amplitudes between 5h and 22h local time obtained from the seismometer and the accelerometer exhibit similar trends and behaviour.

Line 105: Please, provide an explanation of why Saturday is the noisier day of the week, according to 4a. Is it due to the of people in the park? But Sunday is equivalent to Friday. Some explanation is in line 214. Just call attention to that.

Line 106: This reduction on the weekends, are you talking about the figs 4a and 4b? In 4c it is not easily identified.

Line 106: Please write the date of the Lockdown, beginning and end?

Line 110: Is it possible to indicate the park in figure 1?

Line 111: Sorry if I misunderstood the sentence but how does a curfew between 22 h and 5 h reduce amplitudes between 5 h and 22 h?

Line 116: Website of the Ministry.

Line 118: Please, provide further explanation of the Re. What does it mean and how is it measured?

Line 120: What is ICOVID? An intitution?

Line 124: How is the Apple data measured? Change in relation to what?

Line 130: Could you plot the airport location in figure 1?

Line 139: Is there a website where we can find the ppsd (Probabilistic Power Spectral Densities) from all stations?

Line 139: Were you able to identify some decrease during the holidays? Like Christmas, Good Friday.

Line 145: Just call the attention of the reader that the average amplitude is different for each station. Like MT18 is 15 nm and MT16 is 1.5 nm?

Line 159: Just to make it easier for the reader, please provide the information about the approximate distance between urban and rural stations.

Line 169: This peak is not so clear in the for the MT18. Some explanation for that? By eye, I believe the MT18 and Re correlation is worst, am I right? Same for Apple's data

Line 174: Where is the airport?

Line 175: I am sorry, I believe the is a problematic sentence: Is there any confirmation by the government that Lockdown 2 was responsible for the mitigation?

Line 176: Please, explain the five phases. Just a short sentence is enough.

Line 188: Please, make reference that the reader can see where those cities are located in Chile looking at Fig 1.

Line 195: CCSP shows an average of 30 nm "noisier" the MT18. some explanation?

Line 197: MG01 shows a strange pattern. Like a strong decrease in January.

Line 231: transmission of virus to transmission of the virus

Line 272: implemented in other high density cities to implemented in other high-density cities

Line 272: Please, cite some examples where we can find those networks and the impact of their study not just in the mobility but as a tool to teach seismology to school students and the society like Barcelona for example.

https://doi.org/10.3389/feart.2020.00009

Figure 1: I would recommend decreasing the coastline thickness, it is blending with the stations

Figure 2: Is it just an impression or the gaps for the 1-10 Hz are narrower than the others? Just some "illusion" played by the colours?

Figure 3: End of the lockdown 2?

Figure 3: Difference between the blue and white background.

---

## Author Comment (AC1) · 30 Mar 2021

All line numbers added in this reply refer to line numbers in the updated "clean" manuscript i.e. that without the track changes.

**1  General comment**

**This paper looks at the links between seismic ambient noise (ASN) recorded
in Chile and the implementation and reduction of mobility restrictions imposed
due to the Covid-19 pandemic. The work looks at both temporal variations and
differences within Chile. An interesting link between the ASN and the "R" value
is suggested; it will be interesting to understand how this association may be
manifested in other countries with different working/commuting patterns. The
strength of this link may be over-stated in this pre-print, but it is worth pursuing,
as the authors point out. I raise only queries and technical points for the authors
to address, and look forward to them making the needed minor adjustments.**

Thank you for your comments and suggestions that improved our manuscript. We tried
to answer all of them.

**2  Queries**

1. **You link ASN and Re. For the readers who are unused to looking at Re,
   could you indicate whether the Re timeseries you plot are thought to be a
   lagging indicator (ie does the value calculated refer to infection on a partic-
   ular date, or does it refer to detection on a particular date and therefore lag
   when infection actually took place.)**

   The effective reproductive number (Re) is an important indicator to detect
   changes in the virus transmission over time, their estimation is used to evalu-
   ate the policies implemented or population immunity. Re represents the average
   number of secondary cases generated by a primary case during the pandemic.
   One of the challenges is to monitor this parameter in near real-time, because it
   depends on the uniformity in the case-reporting protocols in each country. However, the Re timeseries plotted correspond to the real-time in which the infection took place.

We complemented section 2.3 properly describing the Re indicator (L90), as well as a brief description of how they estimate this parameter:

"The Re indicator is defined as the actual average number of secondary cases generated by a primary case during the epidemic outbreak (Caicedo-Ochoa et al., 2020; Tariq et al., 2021), their estimation is helpful to the assessment of public policies, to estimate population immunity, to monitor near real-time changes in transmission of the viruses over time, among others (Gostic et al., 2020). To control an epidemic outbreak, the Re indicator needs to be reduced below one (Riley et al., 2003). Herein, we used the estimation provided by ICOVID Chile (2020) who described the function Re depending on the proportion of susceptible individuals to be infected, a transmission coefficient and the infectious life expectancy. In other words, the Re accounts for the coefficient between the new infections and the recovery rates plus mortality rates (Contreras et al., 2020). ICOVID Chile (2020) used the method proposed by Cori et al. (2013) to monitor Re in real-time, modelling the transmission like a Poisson process calculated on the basis of the last seven days. We considered only the Re median and 95% credible interval estimated for the urban area in the MR, according to the data given by the Health Service of Santiago City."

New references:

Contreras S, Villavicencio HA, Medina-Ortiz D, Saavedra CP and Olivera-Nappa Á (2020) Real-Time Estimation of Rt for Supporting Public-Health Policies Against COVID-19. Front. Public Health 8:556689. doi: 10.3389/fpubh.2020.556689

Gostic, K. M., McGough, L., Baskerville, E. B., Abbott, S., Joshi, K., Tedijanto, C., ... Cobey, S. (2020). Practical considerations for measuring the effective reproductive number, R t. PLoS computational biology, 16(12), e1008409.

Tariq, A., Undurraga, E. A., Laborde, C. C., Vogt-Geisse, K., Luo, R., Rothenberg, R., Chowell, G. (2021). Transmission dynamics and control of COVID-19 in Chile, March-October, 2020. PLoS neglected tropical diseases, 15(1), e0009070.

2. **Following on from the previous point: a little more description of the Re calculations would be helpful. For example, does the Re cover all of the MR, or a district of the MR containing seismic station MT14. This kind of information would be useful to consider in the context of the limited reach of ASN noted in line (line 46) and the mention of small-area lockdown in line 230.**

Thanks for your comment. We added a description of the Re calculations, and what it means. Regarding your question, the Re plotted in Figure 7 cover only the urban area of the MR (Health Service of Santiago in MR). This includes the area of the eastern Santiago district, where MT14 is located. We added the next sentence in section 2.3 (L99):

"We considered only the Re median and 95% credible interval estimated for the urban area in the MR, according to the data given by the Health Service of Santiago City."

3. **You plot changes in ASN and mobility in figure A4. What are these changes relative to? (ie what is 100%?).**

Thanks for your question. The ASN and Apple mobility data plotted in Figure A4 are relative to a baseline value from 13 January, which corresponds to the first day since Apple public their data. We clarify this in the caption of Figure A4.

"The ASN amplitudes and Apple mobility data are normalised by a baseline value of the 13 January 2020."

4. **Figure 4 shows the MT18 data. Why does it end in August when you have data up to October?**

For MT18 we only included data from 23 January 2020 to 10 August 2020. Unfortunately, this station has considerable data gaps during the months of March and August, but we decide to use their seismic data due to the key location within Santiago City. The data ends in August because the Centro Sismológico Nacional operators changed the seismic instrument recording in the location of MT18.

5. **Lines 108-112 are results, not methods. Consider moving them to the appropriate section.**

   We agree with your comment and move this paragraph to the new subsection 3.1 (L110).

6. **Was lockdown 2 ever lifted? What are phases 2 and 3? it might be good to have a brief paragraph explaining these (somewhere before the results section would seem to be appropriate). Some of this material is already present in lines 176-184.**

   Thanks for this comment and suggestion, we added the new Table A1 that includes more information explaining the five different phases included in the step-by-step programme.

   In addition, we added the next line in the Introduction section (L21):

   "During this first period, the main public health policy addresses the isolation and social distance, including the closure of schools, universities and other educational centres (16 March), national night-time curfew (23 March), and the lockdown of communes. From 19 July 2020, the Chilean government implemented the step-by-step programme, which considers a gradual open of each commune by five phases, based on the monitoring of epidemiological and health system indicators (see Table A1; Tariq et al., 2021)."

   Besides, we include the new Table A2, which remarks the days in which the different cities analysed move from lockdown to phases.

Regarding your question, Lockdown 2 was lifted when "Phase 2: Transition" was applied.

7. **You report a "strong correlation" (line 167) between Re and ASN at station MT14. Correlation is often used in a mathematical sense. Do you have a mathematical relationship in mind here, or are you looking more at matching patterns?**

   We agree with your comment, we consider more a matching pattern than a "strong correlation". We replace "strong correlation" by "matching pattern" statement in the manuscript (L151).

8. **The link between Re and ASN is stronger before 'phase 2' than after it. Is there any reason for this? Might there be some ASN generating activities which are not linked to changes in Re? (I am not an epidemiologist, and this is not an epidemiological paper, but at least acknowledging that the relationship between Re and ASN changes seems to be appropriate).**

   Yes, we think that the better match between Re and ASN before Phase 2 can be explained by the worst implementation of public health policy in those dates, including an early deconfinement in mid-April. Re indicator can assess the effectiveness of the different policies implemented to manage the epidemic outbreak, so we think that during the best matching pattern, the Re peak reflects only the high mobility within city without health cares. After mid-July, the implementation of the step-by-step strategy which proposed a gradual opening according to periodic monitoring of epidemiological and health system indicators reinforced the social distancing interventions and therefore, slowed the spread of the virus (Tariq et al., 2021).

   We added the next sentence in the section "Discussions" (L216) "Although the ASN amplitudes increased due to Phase 2 and Phase 3 of deconfinement in eastern MR, the Re parameter was not linked, indicating better management of

the epidemic outbreak with the broad-scale social distancing interventions implemented in MR (Tariq et al, 2021)."

9. **The paper already mentioned other work in other countries, but I would appreciate a brief paragraph which let me know if the links between ASN and other observables are comparable to, or stronger or weaker than, other metropolitan areas. Does the MR look like Barcelona or Mexico City or Rio de Janeiro or Auckland (this may be beyond the scope of the paper and in that case the authors should feel free to ignore this comment).**

Thanks for your comment. We took your suggestion and we added a brief paragraph about the ASN variability and other observables (mobility data) in other cities-countries, as a reference for the potential readers (see L35).

"Previous works in other countries compare the temporal variability between ASN and other observables such as mobility data from cell phone displacements in northern Italy (Poli et al., 2020), Río de Janeiro, Brazil (Dias et al., 2020), Sicily, Italy (Canatta et al., 2021), Auckland, New Zealand (van Wijk et al., 2021), Barcelona, Spain (Díaz et al., 2020), and Querétaro, México (De Plaen et al., 2020). In addition, Xiao et al. (2020) reported cultural noise changes in China, as well as Guenaga et al. (2021) distinguished significant ASN reductions in academic institutions across the United States."

**3  Technical corrections**

1. **The first sentence (line 34-35) would benefit from a reference from the scientific policy literature.**

We agree. We added the next reference (L19):

Walker, P. G., Whittaker, C., Watson, O. J., Baguelin, M., Winskill, P., Hamlet, A.,

... & Ghani, A. C. (2020). The impact of COVID-19 and strategies for mitigation and suppression in low-and middle-income countries. Science, 369(6502), 413-422.

2. **Line 40 – km2 → km2**

   This has been changed. (L28)

3. **Line 46 – anthropic → anthropogenic (we're making the noise).**

   This has been changed. (L31)

4. **Line 92 – to better understand the effects of the chosen corner frequency?**

   Thanks for your comment, this has been changed. (L74)

5. **Lines 108-110 – be clearer about the time windows over which the 'gradual' reduction happens, and when the changes cease.**

   Thanks for your suggestion. We modify the paragraph and move to another section (L110) since we described results instead of methods:

   "3.1 Lockdown, curfew and ASN amplitudes We analysed the seismic effect caused by the first lockdown in Santiago City using the 24-h clock plots in station MT18 (Figure 4a, 4b). Although we observed a gradual reduction in ASN amplitudes on weekdays due to the day-cares, schools and universities near the station closed (16 March), we also notice a strong reduction on weekends, especially between 11h and 19h local time. Figure A1b shows the area close to MT18 in which we can distinguish the hippodrome "Club Hípico de Santiago" and the O'Higgins Park. The highest ASN amplitudes observed on Saturday before Lockdown 1 (Figure 4a) is explained by the activities of the hippodrome on Saturdays (and some Thursdays during January-February). The hippodrome closed on 21 March 2021, which is in agreement with the decrease in the ASN amplitudes observed after Lockdown 1 (Figure 4b).

We also distinguish the lockdown effect in the hourly grid representation (Figure 4c). The large ASN amplitudes observed during holidays are associated with near activities in both hippodrome and O'Higgins Park, which only persist on weekends during March. After the implementation of Lockdown 1, the ASN amplitudes drop, especially on weekends. Moreover, we observed a systematic behaviour of lower ASN amplitudes between 22h and 5h local time due to the overnight curfew implemented at the same hours, imposed from Lockdown 1 and remain during the full time-window studied."

6. **Line 112 – the noise doesn't go back after lockdown 1 lifts – can you comment on this?**

Thanks for your question. Yes, the ASN amplitudes did not go back after LD1 lifts (period 26 March - 13 April), we only can see a slight increase in this period. In addition, the government lifts the Lockdown without any public health policy to decrease the number of positive cases of COVID-19. After the school, universities and other educational centres closed during March, the public opinion maintain was sceptic about the first lift and most non-sciential companies preserved the lockdown status to avoid social mobility. Also, in this period the over-night curfew was maintained (22 pm – 5 am).

7. **Line 124 – "Related to mobility data, we analysed" → "the mobility data we analysed is"**

This has been changed. (L101)

8. **Line 125 – What actually is**

To avoid misunderstanding, we modify the sentence (L101):

"The mobility data we analysed is provided by Apple mobile-phone locations in Santiago City, which corresponds to the percentage of change in the public's walking and driving in relation to a baseline value from 13 January (Apple, 2020)."

9. **Line 128 – could you explain what a mobility card is?**

A mobility card is the only system to make transactions in public transport, is better known as BIP! Card in the Metropolitan region. We complemented the information in the paragraph (L104):

"They account for the total number of validations using the public transportation card in the MR. This mobility card is the only system to make transactions in public transport."

10. **Line 158 – do you know if the local activity is more likely to be anthropogenic or seismogenic, or is it hard to tell from the data available?**

Thanks for your question. We think that at this point is hard to discriminate it from the regional data available. Probably a further study could include more details about the local activity and their relation with anthropogenic activity, but this is out of the scope of our present work.

11. **Line 167 – different to what? (a range of different responses?)**

Yes, with the word "different" we want to include a range of different responses. We modify the sentence (L150) by:

"and the area implemented a diversity of public policies for mitigating the effects of the pandemic".

12. **Line 194 – might benefit from a reference to oceanic seismic noise for the interested reader?**

We agree, and we added the next reference about oceanic seismic noise (L175):

Cessaro, R. K. (1994). Sources of primary and secondary microseisms. Bulletin of the Seismological Society of America, 84(1), 142-148.

Ardhuin, F., Stutzmann, E., Schimmel, M., & Mangeney, A. (2011). Ocean wave sources of seismic noise. Journal of Geophysical Research: Oceans, 116(C9).

13. **Line 280 – which network code is appropriate? And is there a doi for the seismic network which could be cited here (this will help the network operators if they are looking to see which papers use their data).**

Corrected, we added a new sentence in the "Data availability" section with the appropriate reference (L258):

Universidad De Chile. (2013). Red Sismologica Nacional. International Federation of Digital Seismograph Networks. https://doi.org/10.7914/SN/C1

14. **Line 322 – is there a volume + page range for Caicedo-Ochoa et al?**

Yes, and it was corrected

15. **Line 331 – doi or link missing for Cuadrado et al**

Corrected

16. **Line 367 – add the rest of the author list? Not sure what SE editorial policy is.**

Corrected

17. **Line 415 – square → squares**

This has been changed.

18. **Line 428 – you have an otherwise un-defined term in the key (H*Z). I assume it's because you've got two different components used at this location, but it's not explained anywhere. Consider relabelling/explaining.**

Thanks for notice. We explained the H*Z key in the Figure 3 caption:

"Key legend H*Z can be applied for broadband (HHZ) and strong-motion (HNZ) seismic data."

19. **Line 435 – the icons for the school closures, and lockdowns 1  2 are really hard to distinguish between. Maybe color more of the icons? Also, is the after-lockdown 1 clock plot (b) for this whole time window, or just until the end of lockdown 1? Please clarify.**

Thanks for your suggestions, we modify Figure 4 and increase the colour size of the icons representing school closures and lockdowns. Regarding your question. The after-lockdown 1 clock plot considers the whole time-window analysed. To better clarify this, we added the dates in the Figure 4 caption:

"(a) before Lockdown 1 (period 23 Jan.  2020 – 25 Mar.  2020) and (b) after Lockdown 1 (period 26 Mar. 2020 – 10 Aug 2020)"

20. **Line 446 – not sure what ratio means here?**

We modify the sentence to avoid misleading. Now, in the Figure 5 caption we wrote:

"the near 2 km distance from stations"

21. **Lines 76, 78 + others + 471 – localized (or localised) → located**

This has been changed.

22. **Line 24-25 – "Finally, we suggest to consider monitoring in real time the changes in ASN amplitudes to be included in the public policies" think about changing to something like "Finally, we suggest that real-time monitoring of changes in ASN amplitudes should be considered as part of public health monitoring".**

This has been changed in L14:

"Finally, we suggest that real-time changes in ASN amplitudes should be considered as part of public health policy in further protocols in Santiago as well as other high-density cities of the world, as has been useful during the recent pandemic."

---

## Author Comment (AC2) · 30 Mar 2021

All line numbers added in this reply refer to line numbers in the updated "clean" manuscript i.e. that without the track changes.

**1 General comment**

**The paper is well written and straightforward. Although several cases of the noise level decrease due to lockdowns, curfews, quarantines in different places have been documented in several countries during last year, it is welcome to have another example from a big city that correlates the decrease with some social index. I recommend the paper for publication after some review.**

Thank you for your comment and the following suggestions to improve our manuscript.

**2 Queries and technical corrections**

1. **Abstract: Please, consider to mention in the abstract the results with other stations in Chile (section 3.3)**

   Thanks for your suggestion, we added a brief sentence about other stations in Chile (L9):

   "The same results are observed in other cities such as Iquique, La Serena, and Concepción."

2. **Line 24: real time to real-time**

   This has been changed. (L14)

3. **Line 25: high density to High-density**

   This has been changed. (L16)

4. **Line 35: risk of the spreading of the virus to risk of the virus spreading.**

   This has been changed. (L19)

5. **Line 65: Please, consider to include large and long-scale events in big cities. I believe that this is important to stress.**

   This has been changed. (L49)

6. **Line 76: Is there any particular reason for the installation of the stations in the capital?**

   The stations are part of the National Seismological Network, a permanent network that allows monitoring the seismic activity in the Metropolitan Region, the largest region with a higher population along Chile. This network was recently improved by the Centro Sismológico Nacional (CSN) to further monitor seismogenic faults along the Andes mountain belt, which represents a risk exposure to the inhabitants of the Metropolitan Region.

   More about the development of CSN can be found in Barrientos (2018).

   Barrientos, S. (2018). The seismic network of Chile. Seismological Research Letters, 89(2A), 467-474.

7. **Line 76: Please, provide further information on the conditions where those five stations are installed. Any special treatment because they are inside the city?**

   Regarding the conditions of the five stations installed in the city, we cite the work of Leyton et al. (2018a) and Leyton et al. (2018b), who gives more details about the installation and geophysical characterization like velocity profile and therefore soil conditions, that also can be found in the CSN website: evtdb.csn.uchile.cl. We do not know other specific studies about the soil conditions in those stations. Also, we did not apply special treatment to the stations within the Metropolitan Region.

   To give more information in our manuscript, we added the next sentence (L58):

"Further information about geophysical characterization and soil conditions where stations were deployed can be found in Leyton et al., (2018a) and Leyton et al., (2018b)."

8. **Line 80: Although you are using the horizontal components to see the noise decrease, I recommend to perform the noise polarity analyses to check the direction(s) of the noise origin(s). I believe it is important to check with the direction of the park, airport.**

   Thanks for your suggestion. We analysed the horizontal ASN amplitudes for MT14 station (Figure R1). The three-component seismic noise shows similar trends and variation over time, supporting the findings of Lecocq et al. (2020).

   On the other hand, we perform Noise Polarization analysis (Park et al., 1987) to one hour of three-component continuous seismic records for three stations: MT18 (near park and hippodrome), MT05 (the nearest station to the airport), and MT14. Moreover, we calculate over three different days between 15h-16h p.m. (local time):

   - 10 March 2020
   - 31 March 2020
   - 27 April 2020

   We selected these days because in those we see important changes in ASN amplitudes in station MT14.

   These Noise polarization analyses perform the ground motion in a range of frequencies composed principally by Rayleigh waves, propagated in the direction of the P waves. Our preliminary results are presented in Figures R2, R3 and R4.

   For station MT05 (Figure R2), we do not see significant differences between the time period analysed. The major amplitudes come from directions SE (110°-170°) and NW (290°-350°), probably due to that in the other directions we can see

the hills and a tiny population (see Figure A1c). The source generates Rayleigh waves, and it is probably a superficial source due to the 80°-90° of maximum amplitude in their vertical component.

For station MT18 (Figure R3), the noise source come preferably from SW (180°-270°), showing that the noise source could be associated with the hippodrome instead of the activity in the Park. Regarding the vertical component, we distinguish a broad range of sources.

For station MT14 (Figure R4), we observed heterogeneous noise source in their horizontal component. Probably due to the multiple directions of generation of Rayleigh waves. For those stations, we can not see a pattern between each period.

Despite these results, we prefer not to show these preliminary polarization analyses, because we think that is out of the main scope of our work and more importantly, we understand that this analysis deserves further work and discussions, such as ellipticity (Kulesh et al., 2007), rectilinearity (Montalbetti and Kanasewich, 1970), or planarity analysis (Jurkevics, 1988). Furthermore, we think that to obtain a robust conclusion, we need to include different periods of time and maybe daily and continuous data to analyse the noise source over-time. Undoubtedly, these results could be an important issue for the special edition, but we prefer to improve our analysis in future work.

References: Jurkevics, A. (1988). Polarization analysis of three-component array data. Bulletin of the seismological society of America, 78(5), 1725-1743.

Kulesh, M., Diallo, M. S., Holschneider, M., Kurennaya, K., Krüger, F., Ohrnberger, M., Scherbaum, F. (2007). Polarization analysis in the wavelet domain based on the adaptive covariance method. Geophysical Journal International, 170(2), 667-678.

Montalbetti, J. F., Kanasewich, E. R. (1970). Enhancement of teleseismic body

phases with a polarization filter. Geophysical Journal International, 21(2), 119-129.

Park, J., Lindberg, C. R., Vernon III, F. L. (1987). Multitaper spectral analysis of high‐frequency seismograms. Journal of Geophysical Research: Solid Earth, 92(B12), 12675-12684.

9. **Line 83: For all stations we processed to For all stations, we processed**

This has been changed. (L67)

10. **Line 93: Normalized in relation to what? Normalized individually or among then?**

We normalised the seismic RMS amplitudes individually. In Figure 2, we first obtain the RMS displacement for each frequency band, then we clipped the data above the 95th percentile and subtract the minimum value. Finally, we divided by the RMS displacements by the maximum value which is recorded in the period before Lockdown 1 (23 Jan. 2020 – 25 Mar. 2020).

11. **Line 103-104: The median day-time amplitudes between 5h and 22h local time obtained from the seismometer and the accelerometer exhibits similar trends and behaviour to "The median day-time amplitudes between 5h and 22h local time obtained from the seismometer and the accelerometer exhibit similar trends and behaviour".**

This has been changed. (L81)

12. **Line 105: Please, provide an explanation of why Saturday is the noisier day of the week, according to 4a. Is it due to the of people in the park? But Sunday is equivalent to Friday. Some explanation is in line 214. Just call attention to that.**

Thanks for your comment. We added the new sub-section 3.1 on Results, where we answer your question

"3.1 Lockdown, curfew and ASN amplitudes

We analysed the seismic effect caused by the first lockdown in Santiago City using the 24-h clock plots in station MT18 (Figure 4a, 4b). Although we observed a gradual reduction in ASN amplitudes on weekdays due to the day-cares, schools and universities near the station closed (16 March), we also notice a strong reduction on weekends, especially between 11h and 19h local time. Figure A1b shows the area close to MT18 in which we can distinguish the hippodrome "Club Hípico de Santiago" and the O'Higgins Park. The highest ASN amplitudes observed on Saturday before Lockdown 1 (Figure 4a) is explained by the activities of the hippodrome on Saturdays (and some Thursdays during January-February). The hippodrome closed on 21 March 2021, which is in agreement to the decrease in the ASN amplitudes observed after Lockdown 1 (Figure 4b).

We also distinguish the lockdown effect in the hourly grid representation (Figure 4c). The large ASN amplitudes observed during holidays are associated with near activities in both hippodrome and O'Higgins Park, which only persist on weekends during March. After the implementation of Lockdown 1, the ASN amplitudes drop, especially on weekends. Moreover, we observed a systematic behaviour of lower ASN amplitudes between 22h and 5h local time due to the overnight curfew implemented at the same hours, imposed from Lockdown 1 and remain during the full time-window studied."

13. **Line 106: This reduction on the weekends, are you talking about the figs 4a and 4b? In 4c it is not easily identified.**

Yes, we refer only to Figures 4a and 4b clock plots.

14. **Line 106: Please write the date of the Lockdown, beginning and end?**

Thanks for your suggestions, we added a new line according to the comments of Referee 1. Also, we added Table A2 with the dates when Lockdown and other phases started and ended in the different cities where stations are located.

15. **Line 110: Is it possible to indicate the park in figure 1?**

This has been changed. We added the park location in Figure 1, and in the new Figure A1b.

16. **Line 111: Sorry if I misunderstood the sentence but how does a curfew between 22 h and 5 h reduce amplitudes between 5 h and 22 h?**

Thanks for this comment, we were wrong about the order of the hours. We rewrote this sentence (L121):

"Moreover, we observed a systematic behaviour of lower ASN amplitudes between 22h and 5h local time due to the overnight curfew implemented at the same hours, imposed from Lockdown 1 and remain during the full time-window studied"

17. **Line 116: Website of the Ministry.**

Corrected (L88): "Our study integrates epidemiological data available in the website of the Chilean Ministry of Science"

18. **Line 118: Please, provide further explanation of the Re. What does it mean and how is it measured?**

Thanks for your query, we complete section 2.3 in which we qualitatively describe the Re indicator, as well as how ICOVID Chile (2020) measured (L88):

"One of the primary indicators of the spreading of viruses and contagion dynamics is the estimation of the effective reproductive number (hereafter Re) from confirmed positive cases of COVID-19 since the date of the beginning of symptoms. The Re indicator is defined as the actual average number of secondary cases generated by a primary case during the epidemic outbreak (Caicedo-Ochoa et al., 2020; Tariq et al., 2021), their estimation is helpful to the assessment of public policies, to estimate population immunity, to monitor near real-time changes

in transmission of the viruses over time, among others (Gostic et al., 2020). To control an epidemic outbreak, the Re indicator needs to be reduced below one (Riley et al., 2003). Herein, we used the estimation provided by ICOVID Chile (2020) who described the function Re depending on the proportion of susceptible individuals to be infected, a transmission coefficient and the infectious life expectancy. In other words, the Re accounts for the coefficient between the new infections and the recovery rates plus mortality rates (Contreras et al., 2020). ICOVID Chile (2020) used the method proposed by Cori et al. (2013) to monitor Re in real-time, modelling the transmission like a Poisson process calculated on the basis of the last seven days. We considered only the Re median and 95% credible interval estimated for the urban area in the MR, according to the data given by the Health Service of Santiago City."

19. **Line 120: What is ICOVID? An intitution?**

    ICOVID Chile is an initiative created by Universidad de Chile, Pontificia Universidad Católica de Chile and Universidad de Concepción, as a collaboration with the Ministry of Health and the Ministry of Science. Since the first months of the pandemic, they are analysing different key indicators that represent a full view of the health situation caused by the SARS-CoV-2 viruses. Website: https://www.icovidchile.cl/que-es-icovid-chile

20. **Line 124: How is the Apple data measured? Change in relation to what?**

    Apple gives mobility trends data, which is available for a limited time during the COVID-19 pandemic. The data is based on cell phone locations for a diverse range category such as "Public walking" or "Public driving" of each country-city (including Santiago de Chile). They shared this information and the change is relative to the baseline value from 13th January (first day available). We recognise that this mobility data is biased by people who are carrying their phones and use the map applications. We modify the sentence by:

"The mobility data we analysed is provided by Apple mobile-phone locations in Santiago City, which corresponds to the percentage of change in the public's walking and driving in relation to a baseline value from 13 January (Apple, 2020)"

21. **Line 130: Could you plot the airport location in figure 1?**

    This has been changed. We added the airport location in Figure 1, and in the new Figure A1c.

22. **Line 139: Is there a website where we can find the ppsd (Probabilistic Power Spectral Densities) from all stations?**

    Unfortunately, this is not available. The CSN does not provide this information.

23. **Line 139: Were you able to identify some decrease during the holidays? Like Christmas, Good Friday.**

    In general, yes, we can identify the decrease in some important holidays such as Christmas and New Year's Day (especially if those dates break the continuity during the weekdays, e.g., 2018). Also, we notice a decrease in the National Holidays during September, especially in 2019. However, these changes in ASN amplitudes are not easy to visualize at this scale and would need an additional figure to do mention in the manuscript.

24. **Line 145: Just call the attention of the reader that the average amplitude is different for each station. Like MT18 is 15 nm and MT16 is 1.5 nm?**

    Thanks for your question, we did not write too much about the ASN amplitudes and their difference from each station, although we decide to add the next sentence to clarify (L140):

    "Concerning the ASN amplitude variability, we observed that the quieter stations in the urban area of MR correspond to MT05, MT14, MT16 and MT03, stations that are located over hills, unlike the MT18 and MT15 stations which are deployed

in the valley. Despite the ASN present an average amplitude difference between each station, the temporal variations can be observed within Santiago City"

25. **Line 159: Just to make it easier for the reader, please provide the information about the approximate distance between urban and rural stations.**

Corrected. We added the next sentence (L146): "Notice that the rural stations analysed are deployed within the MR at a distance of about 15 km to 60 km from the stations installed in urban areas (see Figure A1)."

26. **Line 169: This peak is not so clear in the for the MT18. Some explanation for that? By eye, I believe the MT18 and Re correlation is worst, am I right? Same for Apple's data**

We agree with your comment, the matching pattern between mobility data, Re and ASN amplitudes in MT18 and other stations within city is poorly constrained. The only comparable station with MT14 could be MT16 which is located in the same municipality (Las Condes). Unfortunately, MT16 has data gaps just in the scope time-window. Regarding your question, we think that this behaviour has a socioeconomic explanation. According to recent studies, the lockdowns in high-income zones (such as Las Condes) have more effectiveness than in low-income zones (such as Santiago and other communes within MR urban area), due to different economical activities, the possibility to work at home, etc. Also, the low-income zones have higher mobility than high-income zones during lockdowns, which can explain why the seismic noise level in station MT18 did not match with the Re indicator.

We explain this in the Discussion section (L220):

"The matching pattern between the mobility data, the Re indicator and the high-frequency ASN amplitudes is well established for the station MT14 located in Las Condes. Nonetheless, this did not occur with other stations in urban areas such as MT18 placed in Santiago downtown. This observation can be further

explained due to the heterogeneity in policy effectiveness against the COVID-19 spread in MR. Bennett (2021) showed that social distancing, quarantines and testing availability are affected by geographical and socioeconomic factors, in which the lockdowns have been more effective in high-income zones (such as Las Condes) rather than lower-income zones (such as the other station analysed in the MR urban area). Furthermore, the people living in high-income zones can reduce their mobility by around 60% while people in low-income zones only reduce their mobility by around 20% during lockdown (Carranza et al., 2020)."

27. **Line 174: Where is the airport?**

The airport is approximately 5 km western of MT05 station. We added the airport location in Figure 1, and in the new Figure A1c.

28. **Line 175: I am sorry, I believe the is a problematic sentence: Is there any confirmation by the government that Lockdown 2 was responsible for the mitigation?**

We agree and changed the sentence to the following (L158):

"After Lockdown 2, in mid-July, the Chilean government proposed the step-by-step programme to mitigate the propagation of SARS-CoV-2 virus towards a gradual re-opening and increase mobility in different counties as a public health policy (see Table A1)."

29. **Line 176: Please, explain the five phases. Just a short sentence is enough.**

Thanks for this comment, we added Table A1 that includes more information explaining the step-by-step programme with the five phases

In addition, we added the next line in the Introduction section (L21):

"During this first period, the main public health policy addresses the isolation and social distance, including the closure of schools, universities and other educational centres (16 March), national night-time curfew (23 March), and the lock-down of communes. From 19 July 2020, the Chilean government implemented the step-by-step programme, which considers a gradual open of each commune by five phases, based on the monitoring of epidemiological and health system indicators (see Table A1; Tariq et al. (2021))."

30. **Line 188: Please, make reference that the reader can see where those cities are located in Chile looking at Fig 1.**

   Thanks, we mention now the Figure 1 (L167) "High-frequency ASN changes were also recorded in other cities along Chile (Figure 1)."

31. **Line 195: CCSP shows an average of 30 nm "noisier" the MT18. some explanation?**

   We added the next sentence to explain the ASN amplitude observed (L177):

   "The ASN amplitudes in Concepción are at least 30 nm noisier than Santiago (MT18), which could be explained by their location on residential areas, but also the different soil conditions where the stations were installed"

32. **Line 197: MG01 shows a strange pattern. Like a strong decrease in January.**

   We agree, the strange pattern could be explained by the station located near an airfield in Puerto Williams. The airport operates on different days a week, probably explaining the strange patterns in seismic noise levels.

   We added the next paragraph (L181):

   "Nevertheless, this station shows a strange pattern before Lockdown. The first one corresponds to a high drop in mid-December until the first days of January associated with holidays festivities (Christmas and New Year Day). The second pattern observed is the temporal variability that could be associated with the activity of the airfield near the site where the station is operating. Unfortunately,

we don't have access to the aeroplane activity in those weeks to support our assumption."

33. **Line 231: transmission of virus to transmission of the virus**

This has been changed. (L208)

34. **Line 272: implemented in other high density cities to implemented in other high-density cities**

This has been changed. (L255)

35. **Line 272: Please, cite some examples where we can find those networks and the impact of their study not just in the mobility but as a tool to teach seismology to school students and the society like Barcelona for example. https://doi.org/10.3389/feart.2020.00009**

Thanks for your recommendation. We agree with it, and we added the next sentence (L252):

"These seismometers are typically used for the management of seismological networks in urban areas; however, recent studies show the potential opportunity to use them as a tool to teach seismology to school students (e.g., Subedi et al., 2020) and increase the interest of society toward Earth Sciences (e.g., Diaz et al., 2020)."

36. **Figure 1: I would recommend decreasing the coastline thickness, it is blending with the stations** This has been changed

37. **Figure 2: Is it just an impression or the gaps for the 1-10 Hz are narrower than the others? Just some "illusion" played by the colours?**

Yes, the data gaps are the same for each frequency analysed.

38. **Figure 3: End of the lockdown 2?**

After Lockdown 2, the government implemented the program step-by-step which include five different phases mentioned in the new Table A1. This program was implemented in mid-July. Since then, the Lockdown 2 lifts on 17 August 2020 (Phase 2). Now, we include these dates in Table A2.

39. **Figure 3: Difference between the blue and white background.**

Corrected, we added the next sentence in Figure 3 caption:

"The grey and white background correspond to weekdays and weekends, respectively."

**3   Figure captions**

- Figure R1. ASN amplitudes for MT14 station in their components: (a) HHZ, (b) HHN, (c) HHE.

- Figure R2. Polarization analysis for station MT05. (a) On 10 March 2020 at 15h-16h local time, (b) on 31 March 2020 at 15h-16h local time, and (c) on 27 April 2020 at 15h-16h local time. The left panel show horizontal variation and the right panel shows vertical variation in theta. If we considered that the noise sources come from P and superficial waves, we can interpret that the horizontal angle ($\theta_H$) is defined by $0° = N$, $90° = E$, $180° = S$, $270° = W$, and the vertical angle ($\theta_V$) in which we can infer $0°$ from deep sources and $90°$ from shallower sources.

- Figure R3. Polarization analysis for station MT14. (a) On 10 March 2020 at 15h-16h local time, (b) on 31 March 2020 at 15h-16h local time, and (c) on 27 April 2020 at 15h-16h local time.

- Figure R4. Polarization analysis for station MT18. (a) On 10 March 2020 at 15h-16h local time, (b) on 31 March 2020 at 15h-16h local time, and (c) on 27 April 2020 at 15h-16h local time.
* * *
[Figure]

**Fig. 1.**

(a) Polarization analysis for MT05 on 10 Mar 2020 (pre-LD1) at 15h - 16h (local time)

(b) Polarization analysis for MT05 on 31 Mar 2020 (post-LD1) at 15h - 16h (local time)

(c) Polarization analysis for MT05 on 28 Apr 2020 (LD2) at 15h - 16h (local time)

**Fig. 2.**

(a) Polarization analysis for MT18 on 10 Mar 2020 (pre-LD1) at 15h - 16h (local time)

(b) Polarization analysis for MT18 on 31 Mar 2020 (post-LD1) at 15h - 16h (local time)

(c) Polarization analysis for MT18 on 28 Apr 2020 (LD2) at 15h - 16h (local time)

**Fig. 3.**

[Figure]

(a) Polarization analysis for MT14 on 10 Mar 2020 (pre-LD1) at 15h - 16h (local time)

(b) Polarization analysis for MT14 on 31 Mar 2020 (post-LD1) at 15h - 16h (local time)

(c) Polarization analysis for MT14 on 28 Apr 2020 (LD2) at 15h - 16h (local time)

**Fig. 4.**

[Figure]

---

## Author Response (AR1)

**Author's response for "Seismic noise variability as an indicator of urban mobility during COVID-19 pandemic in Santiago Metropolitan Region, Chile"**

Javier Ojeda and Sergio Ruiz

Departamento de Geofísica, Universidad de Chile

March 30, 2021

Firstly, we would like to thank the reviewers for their fruitful comments and suggestions that they made to improve our work. We agree with most of them and worked on it to make clear our study.

Most of the point-by-point response presented here was included in the interactive comments (Reply on RC1 and Reply on RC2). However, we sort some of them according to the track changes of our manuscript.

All line numbers added in this reply refer to line numbers in the updated clean manuscript.

**1 Reviewer 1**

1. **Regarding the Re timeseries, description and calculations**

   We complemented section 2.3 properly describing the Re indicator (L90), as well as a brief description of how ICOVID Chile (2020) estimate this parameter:

   "The Re indicator is defined as the actual average number of secondary cases generated by a primary case during the epidemic outbreak (Caicedo-Ochoa et al., 2020; Tariq et al., 2021), their estimation is helpful to the assessment of public policies, to estimate population immunity, to monitor near real-time changes in transmission of the viruses over time, among others (Gostic et al., 2020). To control an epidemic outbreak, the Re indicator needs to be reduced below one (Riley et al., 2003). Herein, we used the estimation provided by ICOVID Chile (2020) who described the function Re depending on the proportion of susceptible individuals to be infected, a transmission coefficient and the infectious life expectancy. In other words, the Re accounts for the coefficient between the new infections and the recovery rates plus mortality rates (Contreras et al., 2020). ICOVID Chile (2020) used the method proposed by Cori et al. (2013) to monitor Re in real-time, modelling the transmission like a Poisson process calculated on the basis of the last seven days. We considered only the Re median and 95% credible interval estimated for the urban area in the MR, according to the data given by the Health Service of Santiago City."

   New references:

   Contreras S, Villavicencio HA, Medina-Ortiz D, Saavedra CP and Olivera-Nappa Á (2020) Real-Time Estimation of Rt for Supporting Public-Health Policies Against COVID-19. Front. Public Health 8:556689. doi: 10.3389/fpubh.2020.556689

   Gostic, K. M., McGough, L., Baskerville, E. B., Abbott, S., Joshi, K., Tedijanto, C., ... & Cobey, S. (2020). Practical considerations for measuring the effective reproductive number, R t. PLoS computational biology, 16(12), e1008409.

   Tariq, A., Undurraga, E. A., Laborde, C. C., Vogt-Geisse, K., Luo, R., Rothenberg, R., & Chowell, G. (2021). Transmission dynamics and control of COVID-19 in Chile, March-October, 2020. PLoS neglected tropical diseases, 15(1), e0009070.

2. **Figure A4. What is 100%?**

   We clarify this adding a brief sentence in the caption of Figure A4.

   "The ASN amplitudes and Apple mobility data are normalised by a baseline value of the 13 January 2020."

3. **Lines 108-112 are results, not methods.**

   We moved this paragraph to the new subsection 3.1 (L110).

4. **Was lockdown 2 ever lifted? What are phases 2 and 3?**

   We added the next line in the Introduction section (L21):

   "During this first period, the main public health policy addresses the isolation and social distance, including the closure of schools, universities and other educational centres (16 March), national night-time curfew (23 March), and the lockdown of communes. From 19 July 2020, the Chilean government implemented the step-by-step programme, which considers a gradual open of each commune by five phases, based on the monitoring of epidemiological and health system indicators (see Table A1; Tariq et al., 2021)."

   Besides, we include the new Table A2, which remarks the days in which the different cities analysed move from lockdown to phases.

5. **You report a "strong correlation" (line 167) between Re and ASN at station MT14**

   We agree with your comment, we consider more a matching pattern than a "strong correlation". We replace "strong correlation" by "matching pattern" statement in the manuscript (L151).

6. **The link between Re and ASN is stronger before 'phase 2' than after it. Is there any reason for this? Might there be some ASN generating activities which are not linked to changes in Re?**

   Regarding this point, we added the next sentence in the section "Discussions" (L216)

   "Although the ASN amplitudes increased due to Phase 2 and Phase 3 of deconfinement in eastern MR, the Re parameter was not linked, indicating better management of the epidemic outbreak with the broad-scale social distancing interventions implemented in MR (Tariq et al, 2021))."

7. **The paper already mentioned other work in other countries, but I would appreciate a brief paragraph which let me know if the links between ASN and other observables are comparable to**

   We included a brief paragraph in L35:

   "Previous works in other countries compare the temporal variability between ASN and other observables such as mobility data from cell phone displacements in northern Italy (Poli et al., 2020), Río de Janeiro, Brazil (Dias et al., 2020), Sicily, Italy (Canatta et al., 2021), Auckland, New Zealand (van Wijk et al., 2021), Barcelona, Spain (Díaz et al., 2020), and Querétaro, México (De Plaen et al., 2020). In addition, Xiao et al. (2020) reported cultural noise changes in China, as well as Guenaga et al. (2021) distinguished significant ASN reductions in academic institutions across the United States."

8. **The first sentence (line 34-35) would benefit from a reference from the scientific policy literature.**

   We added the next reference (L19):

   Walker, P. G., Whittaker, C., Watson, O. J., Baguelin, M., Winskill, P., Hamlet, A., ... & Ghani, A. C. (2020). The impact of COVID-19 and strategies for mitigation and suppression in low-and middle-income countries. Science, 369(6502), 413-422.

9. **Line 40 – km2 → km2**

   This has been changed. (L28)

10. **Line 46 – anthropic → anthropogenic (we're making the noise).**

    This has been changed. (L31)

11. **Line 92 – to better understand the effects of the chosen corner frequency?**

    This has been changed. (L74)

12. **Lines 108-110 – be clearer about the time windows over which the 'gradual' reduction happens, and when the changes cease.**

    We modified the paragraph and moved to another section (L110) since we described results instead of methods:

    "3.1 Lockdown, curfew and ASN amplitudes

    We analysed the seismic effect caused by the first lockdown in Santiago City using the 24-h clock plots in station MT18 (Figure 4a, 4b). Although we observed a gradual reduction in ASN amplitudes on weekdays due to

the day-cares, schools and universities near the station closed (16 March), we also notice a strong reduction on weekends, especially between 11h and 19h local time. Figure A1b shows the area close to MT18 in which we can distinguish the hippodrome "Club Hípico de Santiago" and the O'Higgins Park. The highest ASN amplitudes observed on Saturday before Lockdown 1 (Figure 4a) is explained by the activities of the hippodrome on Saturdays (and some Thursdays during January-February). The hippodrome closed on 21 March 2021, which is in agreement with the decrease in the ASN amplitudes observed after Lockdown 1 (Figure 4b).

We also distinguish the lockdown effect in the hourly grid representation (Figure 4c). The large ASN amplitudes observed during holidays are associated with near activities in both hippodrome and O'Higgins Park, which only persist on weekends during March. After the implementation of Lockdown 1, the ASN amplitudes drop, especially on weekends. Moreover, we observed a systematic behaviour of lower ASN amplitudes between 22h and 5h local time due to the overnight curfew implemented at the same hours, imposed from Lockdown 1 and remain during the full time-window studied."

13. **Line 124 – "Related to mobility data, we analysed" → "the mobility data we analysed is"**

   This has been changed. (L101)

14. **Line 125 – What actually is. Mobility data**

   We modified the sentence (L101):

   "The mobility data we analysed is provided by Apple mobile-phone locations in Santiago City, which corresponds to the percentage of change in the public's walking and driving in relation to a baseline value from 13 January (Apple, 2020)."

15. **Line 128 – could you explain what a mobility card is?**

   We complemented the information in the paragraph (L104):

   "They account for the total number of validations using the public transportation card in the MR. This mobility card is the only system to make transactions in public transport."

16. **Line 167 – different to what? (a range of different responses?)**

   We modified the sentence (L150) by:

   "and the area implemented a diversity of public policies for mitigating the effects of the pandemic".

17. **Line 194 – might benefit from a reference to oceanic seismic noise for the interested reader?**

   We added the next reference about oceanic seismic noise (L175):

   Cessaro, R. K. (1994). Sources of primary and secondary microseisms. Bulletin of the Seismological Society of America, 84(1), 142-148.

   Ardhuin, F., Stutzmann, E., Schimmel, M., & Mangeney, A. (2011). Ocean wave sources of seismic noise. Journal of Geophysical Research: Oceans, 116(C9).

18. **Line 280 – which network code is appropriate? And is there a doi for the seismic network which could be cited here.**

   Corrected, we added a new sentence in the "Data availability" section with the appropriate reference (L258):

   Universidad De Chile. (2013). Red Sismologica Nacional. International Federation of Digital Seismograph Networks. https://doi.org/10.7914/SN/C1

19. **Line 322 – is there a volume + page range for Caicedo-Ochoa et al?**

   Yes, and it was corrected

20. **Line 331 – doi or link missing for Cuadrado et al**

   Corrected

21. **Line 367 – add the rest of the author list? Not sure what SE editorial policy is.**

   Corrected

22. **Line 415 – square → squares**

    This has been changed. (Figure 1 caption)

23. **Line 428 – you have an otherwise un-defined term in the key (H\*Z). I assume it's because you've got two different components used at this location, but it's not explained anywhere. Consider rela-belling/explaining.**

    We explained the H\*Z key in the Figure 3 caption:

    "Key legend H\*Z can be applied for broadband (HHZ) and strong-motion (HNZ) seismic data."

24. **Line 435 – the icons for the school closures, and lockdowns 1  2 are really hard to distinguish be-tween. Maybe color more of the icons?**

    We modified Figure 4 and increase the colour size of the icons representing school closures and lockdowns.

25. **Is the after-lockdown 1 clock plot (b) for this whole time window, or just until the end of lockdown 1? Please clarify**

    We added the dates in the Figure 4 caption:

    "(a) before Lockdown 1 (period 23 Jan. 2020 – 25 Mar. 2020) and (b) after Lockdown 1 (period 26 Mar. 2020 – 10 Aug 2020)"

26. **Line 446 – not sure what ratio means here?**

    We modified the sentence to avoid misleading. Now, in the Figure 5 caption we wrote:

    "the near 2 km distance from stations"

27. **Lines 76, 78 + others + 471 – localized (or localised) → located**

    This has been changed, and some of them were changed by "placed"

28. **Line 24-25 – "Finally, we suggest to consider monitoring in real time the changes in ASN amplitudes to be included in the public policies" think about changing to something like "Finally, we suggest that real-time monitoring of changes in ASN amplitudes should be considered as part of public health monitoring".**

    This has been changed in L14:

    "Finally, we suggest that real-time changes in ASN amplitudes should be considered as part of public health policy in further protocols in Santiago as well as other high-density cities of the world, as has been useful during the recent pandemic."

**2 Reviewer 2**

1. **Abstract: Please, consider to mention in the abstract the results with other stations in Chile (section 3.3)**

   We added a brief sentence about other stations in Chile (L9):

   "The same results are observed in other cities such as Iquique, La Serena, and Concepción."

2. **Line 24: real time to real-time**

   This has been changed. (L14)

3. **Line 25: high density to High-density**

   This has been changed. (L16)

4. **Line 35: risk of the spreading of the virus to risk of the virus spreading.**

   This has been changed. (L19)

5. **Line 65: Please, consider to include large and long-scale events in big cities. I believe that this is important to stress.**

   This has been changed. (L49)

6. **Line 76: Please, provide further information on the conditions where those five stations are installed. Any special treatment because they are inside the city?**

   To give more information in our manuscript, we added the next sentence (L58):

   "Further information about geophysical characterization and soil conditions where stations were deployed can be found in Leyton et al., (2018a) and Leyton et al., (2018b)."

7. **Line 83: For all stations we processed to For all stations, we processed**

   This has been changed. (L67)

8. **Line 103-104: The median day-time amplitudes between 5h and 22h local time obtained from the seismometer and the accelerometer exhibits similar trends and behaviour to "The median day-time amplitudes between 5h and 22h local time obtained from the seismometer and the accelerometer exhibit similar trends and behaviour".**

   This has been changed. (L81)

9. **Line 105: Please, provide an explanation of why Saturday is the noisier day of the week, according to 4a. Is it due to the of people in the park? But Sunday is equivalent to Friday. Some explanation is in line 214. Just call attention to that.**

   We added the new sub-section 3.1 on Results, where we answer your question

   "3.1 Lockdown, curfew and ASN amplitudes

   We analysed the seismic effect caused by the first lockdown in Santiago City using the 24-h clock plots in station MT18 (Figure 4a, 4b). Although we observed a gradual reduction in ASN amplitudes on weekdays due to the day-cares, schools and universities near the station closed (16 March), we also notice a strong reduction on weekends, especially between 11h and 19h local time. Figure A1b shows the area close to MT18 in which we can distinguish the hippodrome "Club Hípico de Santiago" and the O'Higgins Park. The highest ASN amplitudes observed on Saturday before Lockdown 1 (Figure 4a) is explained by the activities of the hippodrome on Saturdays (and some Thursdays during January-February). The hippodrome closed on 21 March 2021, which is in agreement to the decrease in the ASN amplitudes observed after Lockdown 1 (Figure 4b).

   We also distinguish the lockdown effect in the hourly grid representation (Figure 4c). The large ASN amplitudes observed during holidays are associated with near activities in both hippodrome and O'Higgins Park, which only persist on weekends during March. After the implementation of Lockdown 1, the ASN amplitudes drop, especially on weekends. Moreover, we observed a systematic behaviour of lower ASN amplitudes between 22h and 5h local time due to the overnight curfew implemented at the same hours, imposed from Lockdown

1 and remain during the full time-window studied."

10. **Line 106: Please write the date of the Lockdown, beginning and end?**

We added Table A2 with the dates when Lockdown and other phases started and ended in the different cities where stations are located.

11. **Line 110: Is it possible to indicate the park in figure 1?**

This has been changed. We added the park location in Figure 1, and in the new Figure A1b.

12. **Line 111: Sorry if I misunderstood the sentence but how does a curfew between 22 h and 5 h reduce amplitudes between 5 h and 22 h?**

We re-wrote this sentence (L121):

"Moreover, we observed a systematic behaviour of lower ASN amplitudes between 22h and 5h local time due to the overnight curfew implemented at the same hours, imposed from Lockdown 1 and remain during the full time-window studied"

13. **Line 116: Website of the Ministry.**

Corrected (L88):

"Our study integrates epidemiological data available in the website of the Chilean Ministry of Science"

14. **Line 118: Please, provide further explanation of the Re. What does it mean and how is it measured?**

We complete section 2.3 in which we qualitatively describe the Re indicator, as well as how ICOVID Chile (2020) measured (L88):

"One of the primary indicators of the spreading of viruses and contagion dynamics is the estimation of the effective reproductive number (hereafter Re) from confirmed positive cases of COVID-19 since the date of the beginning of symptoms. The Re indicator is defined as the actual average number of secondary cases generated by a primary case during the epidemic outbreak (Caicedo-Ochoa et al., 2020; Tariq et al., 2021), their estimation is helpful to the assessment of public policies, to estimate population immunity, to monitor near real-time changes in transmission of the viruses over time, among others (Gostic et al., 2020). To control an epidemic outbreak, the Re indicator needs to be reduced below one (Riley et al., 2003). Herein, we used the estimation provided by ICOVID Chile (2020) who described the function Re depending on the proportion of susceptible individuals to be infected, a transmission coefficient and the infectious life expectancy. In other words, the Re accounts for the coefficient between the new infections and the recovery rates plus mortality rates (Contreras et al., 2020). ICOVID Chile (2020) used the method proposed by Cori et al. (2013) to monitor Re in real-time, modelling the transmission like a Poisson process calculated on the basis of the last seven days. We considered only the Re median and 95% credible interval estimated for the urban area in the MR, according to the data given by the Health Service of Santiago City."

15. **Line 124: How is the Apple data measured? Change in relation to what?**

We modified the sentence (L101) by:

"The mobility data we analysed is provided by Apple mobile-phone locations in Santiago City, which corresponds to the percentage of change in the public's walking and driving in relation to a baseline value from 13 January (Apple, 2020)"

16. **Line 130: Could you plot the airport location in figure 1?**

This has been changed. We added the airport location in Figure 1, and in the new Figure A1c.

17. **Line 145: Just call the attention of the reader that the average amplitude is different for each station. Like MT18 is 15 nm and MT16 is 1.5 nm?**

We added the next sentence (L140):

"Concerning the ASN amplitude variability, we observed that the quieter stations in the urban area of MR correspond to MT05, MT14, MT16 and MT03, stations that are located over hills, unlike the MT18 and MT15 stations which are deployed in the valley. Despite the ASN present an average amplitude difference between

each station, the temporal variations can be observed within Santiago City"

18. **Line 159: Just to make it easier for the reader, please provide the information about the approximate distance between urban and rural stations.**

We added the next sentence (L146):

"Notice that the rural stations analysed are deployed within the MR at a distance of about 15 km to 60 km from the stations installed in urban areas (see Figure A1)."

19. **Line 169: This peak is not so clear in the for the MT18. Some explanation for that? By eye, I believe the MT18 and Re correlation is worst, am I right? Same for Apple's data**

We added a new paragraph to the Discussion section (L220):

"The matching pattern between the mobility data, the Re indicator and the high-frequency ASN amplitudes is well established for the station MT14 located in Las Condes. Nonetheless, this did not occur with other stations in urban areas such as MT18 placed in Santiago downtown. This observation can be further explained due to the heterogeneity in policy effectiveness against the COVID-19 spread in MR. Bennett (2021) showed that social distancing, quarantines and testing availability are affected by geographical and socioeconomic factors, in which the lockdowns have been more effective in high-income zones (such as Las Condes) rather than lower-income zones (such as the other station analysed in the MR urban area). Furthermore, the people living in high-income zones can reduce their mobility by around 60% while people in low-income zones only reduce their mobility by around 20% during lockdown (Carranza et al., 2020)."

20. **Line 175: I am sorry, I believe the is a problematic sentence: Is there any confirmation by the government that Lockdown 2 was responsible for the mitigation?**

We modified the sentence by (L158):

"After Lockdown 2, in mid-July, the Chilean government proposed the step-by-step programme to mitigate the propagation of SARS-CoV-2 virus towards a gradual re-opening and increase mobility in different counties as a public health policy (see Table A1)."

21. **Line 176: Please, explain the five phases. Just a short sentence is enough.**

We added the new Table A1 that includes more information explaining the step-by-step programme with the five phases

In addition, we added the next line in the Introduction section (L21):

"During this first period, the main public health policy addresses the isolation and social distance, including the closure of schools, universities and other educational centres (16 March), national night-time curfew (23 March), and the lockdown of communes. From 19 July 2020, the Chilean government implemented the step-by-step programme, which considers a gradual open of each commune by five phases, based on the monitoring of epidemiological and health system indicators (see Table A1; Tariq et al. (2021))."

22. **Line 188: Please, make reference that the reader can see where those cities are located in Chile looking at Fig 1.**

We added the mention to the Figure 1 (L167):

"High-frequency ASN changes were also recorded in other cities along Chile (Figure 1)."

23. **Line 195: CCSP shows an average of 30 nm "noisier" the MT18. some explanation?**

We added the next sentence to explain the ASN amplitude observed (L177):

"The ASN amplitudes in Concepción are at least 30 nm noisier than Santiago (MT18), which could be explained by their location on residential areas, but also the different soil conditions where the stations were installed"

24. **Line 197: MG01 shows a strange pattern. Like a strong decrease in January.**

We added the next paragraph (L181):

"Nevertheless, this station shows a strange pattern before Lockdown. The first one corresponds to a high drop in mid-December until the first days of January associated with holidays festivities (Christmas and New Year

Day). The second pattern observed is the temporal variability that could be associated with the activity of the airfield near the site where the station is operating. Unfortunately, we do not have access to the aeroplane activity in those weeks to support our assumption."

25. **Line 231: transmission of virus to transmission of the virus**

This has been changed. (L208)

26. **Line 272: implemented in other high density cities to implemented in other high-density cities**

This has been changed. (L255)

27. **Line 272: Please, cite some examples where we can find those networks and the impact of their study not just in the mobility but as a tool to teach seismology to school students and the society like Barcelona for example. https://doi.org/10.3389/feart.2020.00009**

We added the next sentence to the Conclusions section (L252):

"These seismometers are typically used for the management of seismological networks in urban areas; however, recent studies show the potential opportunity to use them as a tool to teach seismology to school students (e.g., Subedi et al., 2020) and increase the interest of society toward Earth Sciences (e.g., Diaz et al., 2020)."

28. **Figure 1: I would recommend decreasing the coastline thickness, it is blending with the stations**

This has been changed in Figure 1 caption.

29. **Figure 3: Difference between the blue and white background.**

We added the next sentence in Figure 3 caption:

"The grey and white background correspond to weekdays and weekends, respectively."

**3  Additional changes**

1. (L45) the Chilean seismic network operated by the National Seismological Centre (hereafter CSN; Centro Sismológico Nacional; Barrientos (2018))

2. (L198) an overnight curfew

3. (L208) Other authors also suggest that small-area lockdowns and reductions in mobility can reduce the transmission of the virus but their impact was smaller than the early closures of schools, universities and other educational centres

4. (L212) The removal of lockdown protocols after 1 May to reopening the economy resulted in a new wave of infections and an exponential increase in the number of positive COVID-19 cases.

5. Figure 7: We changed the size of the figure and we modified the labels for the times of public restrictions implemented (School closures, LD1, end LD1, etc.)

6. Figure A1: We added a close view to the civinity of MT18 and MT05 stations.

7. Figure A4: We changed the size of the figure and we modified the labels for the times of public restrictions implemented (School closures, LD1, end LD1, etc.)

8. Figure A5: We changed the size of the figure and we modified the labels for the times of public restrictions implemented (School closures, LD1, end LD1, etc.)

On behalf of the authors,

Javier Ojeda

---

## Author Response (AR2)

**Author's response for "Seismic noise variability as an indicator of urban mobility during COVID-19 pandemic in Santiago Metropolitan Region, Chile"**

Javier Ojeda and Sergio Ruiz
Departamento de Geofísica, Universidad de Chile
April 8, 2021

Firstly, we would like to thank to Raphael De Plaen (topical Editor) as well as the reviewers for their fruitful comments and suggestions that they made to improve our work.

In the accepted manuscript we made a few change listed here:

1. Instead of an Appendix section at the end of our manuscript, we decided to build a Supplementary Material in a *.pdf file.

2. Due to point 1, we changed all the labels AX by SX in our manuscript (including tables and figures). For example, Table A1 now is labelled as Table S1.

3. In the discussion section, we re-wrote the sentence in L209 by:

   "Also, Cuadrado et al. (2020) suggest that small-area lockdowns and reductions in mobility can reduce the transmission of the virus but their impact was smaller than the early closures of schools, universities and other educational centres."

4. In the discussion section also, we separated in a different paragraph since L216 because of the long extension of that paragraph.

5. Finally, we added the last sentence in L221 to better finalize our discussions:

   "However, according to Canals et al. (2020), a relaxing of these interventions could rise the infections and saturate the health systems."

On behalf of the authors,

Javier Ojeda